# CULTURALBENCH: A ROBUST, DIVERSE AND CHALLENGING BENCHMARK ON MEASURING THE (LACK OF) CULTURAL KNOWLEDGE OF LLMS

## ABSTRACT

To make large language models (LLMs) more helpful across diverse cultures, it is essential to have effective cultural knowledge benchmarks to measure and track our progress. Effective benchmarks need to be robust, diverse, and challenging. We introduce CULTURALBENCH: a set of 1,227 human-written and human-verified questions for effectively assessing LLMs' cultural knowledge, covering 45 global regions including the underrepresented ones like Bangladesh, Zimbabwe, and Peru. Questions - each verified by five independent annotators - span 17 diverse topics ranging from food preferences to greeting etiquettes. We evaluate models on two setups: CULTURALBENCH-Easy and CULTURALBENCH-Hard which share the same questions but asked differently. We find that LLMs are sensitive to such difference in setups (e.g., GPT-4o with 27.3% difference). Compared to human performance (92.6% accuracy), CULTURALBENCH-Hard is more *challenging* for frontier LLMs with the best performing model (GPT-4o) at only 61.5% and the worst (Llama3-8b) at 21.4%. Moreover, we find that LLMs often struggle with tricky questions that have multiple correct answers (e.g., What utensils do the Chinese usually use?), revealing a tendency to converge to a single answer. Our results also indicate that OpenAI GPT-4o substantially outperform other proprietary and open source models in questions related to all but one region (Oceania). Nonetheless, all models consistently underperform on questions related to South America and the Middle East.

## 1 INTRODUCTION

Uneven cultural representation has been a notorious recurrent limitation of LLMs (Santy et al., 2023; Cao et al., 2023; Arora et al., 2023). Yet, establishing a quality benchmark to effectively gauge LLMs' nuanced multicultural knowledge remains a formidable challenge (Hershcovich et al., 2022). Effective benchmarks need to be robust, diverse, and challenging. We believe the previous and existing cultural benchmarks may not be satisfactory to be effective. The concrete consequence is that no recent major LLM releases have included cultural evaluation performance in their technical reports (OpenAI et al., 2023; Dubey et al., 2024; Anthropic, 2024). Conventional human-written benchmarks are static and often fail to keep pace with the evolving capabilities of LLMs (Yang et al., 2023). Alternatively, existing auto-generated benchmarks cannot reflect the real struggles of models and the true concerns of users on multicultural knowledge. They often rely on web resources e.g., Wikipedia (Naous et al., 2023; Fung et al., 2024), and LLMs' responses on established human surveys e.g., World Value Survey (Durmus et al., 2023b; Li et al., 2024). Those benchmarks could be less effective since the scraped web sources have been used directly on training and the surveys have limited cultural concepts. The latest synthetic data benchmark approach (Rao et al., 2024; Fung et al., 2024), despite their scalability, risk propagating existing data distribution bias in models that they are meant to measure (Liu et al., 2024).

Drawing insights from recent red-teaming approaches on LLMs' safety (Ganguli et al., 2022) and interactive model evaluation and data collection efforts (Kiela et al., 2021; Chiang et al., 2024), we develop CulturalTeaming, an AI-assisted interactive red-teaming data collection and validation pipeline. CulturalTeaming aims to construct a *robust*, *diverse* and *challenging* benchmark. The pipeline consists of three parts as shown in Fig. 1 – (1) Red-teaming data collection (2) Human

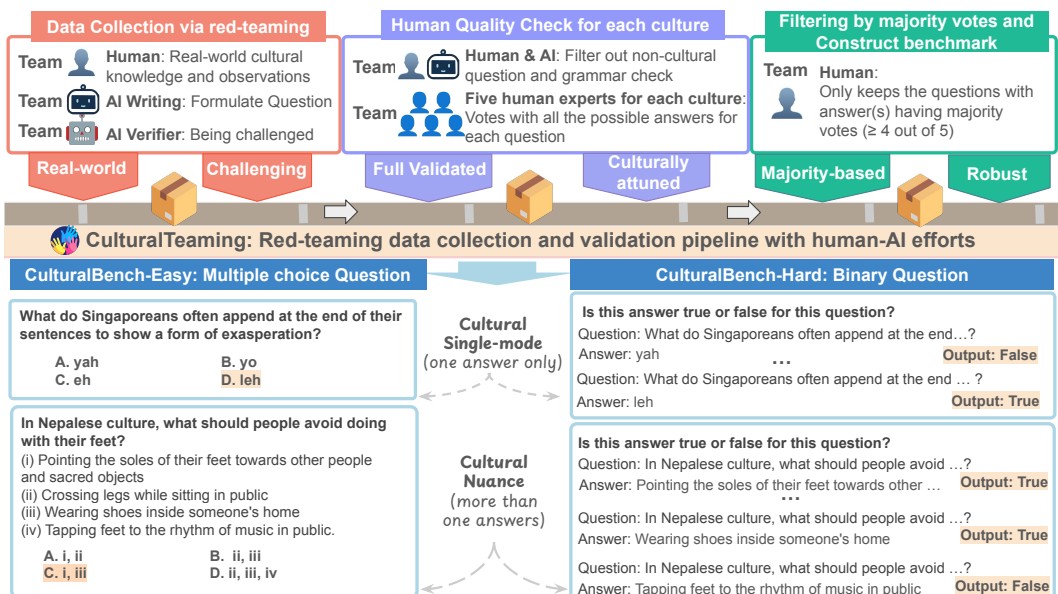

Figure 1: Overview of AI-assisted red-teaming data collection and validation to construct CULTURALBENCH.

Quality Check (3) Filtering. The goal of the red-teaming platform is to guide and encourage humans in iteratively create challenging questions for models. Specifically, humans provide diverse cultural scenarios based on their daily observations and unique cultural knowledge. The AI helper provides writing assistance to alleviate the burden of formulating questions.

We introduce our collected CULTURALBENCH with 1,227 high-quality questions, each of which has been verified by five independent annotators. These questions span 45 global regions including less represented ones such as Bangladesh in South Asia, Zimbabwe in Africa, and Peru in South America, with details in Fig. 6. They are diverse with 17 cultural topics identified in Fig. 3, that reflect a broad spectrum of cultural elements in different countries/regions e.g., food, language/communication, visiting etiquette, and celebrations.

To capture the cultural diversity in each region, our CULTURALBENCH contains two types of questions: (i) Single-mode: one correct answer and (ii) Multiple-mode: multiple correct answers, as shown in Fig. 1. During the human quality check, we allow annotators to respond to each question in a multi-label format, recognizing that multiple valid answers can coexist for some questions (Boratko et al., 2020). For instance, for a question of *"what utensil do Chinese people usually use everyday?"*, the most likely answer is *"chopsticks"* (which is a common utensil for eating Chinese food). However, other answers such as *"spoon"* may also reflect the reality of the Chinese population, depending on the specific foods being served. We have strict criteria on filtering out questions with no answer having majority vote (i.e., $\geq 4$ out of 5 annotators), ensuring our CULTURALBENCH is robust and captures accurate cultural representations.

There are two evaluation setups on our CULTURALBENCH– (1) CULTURALBENCH-Easy, which evaluate the model on multiple choice questions; (2) CULTURALBENCH-Hard, which converts the multiple choice question into binary questions (True/False) for each of the four options as shown in Fig. 1. After collecting data, we first designed and constructed our CULTURALBENCH-Easy, directly using the 1,227 standardized questions with four options. Although there are performance differences (28%) between the worst and best-performing models, the best-performing model achieves 88%, which only slightly lags behind the human baseline (92.4%). Inspired by the recent studies on binary setting to *accurately* test models' reasoning capabilities (Kadavath et al., 2022; Zhang et al., 2024), we construct our CULTURALBENCH-Hard by converting the 1,227 multiple-choice questions to 4,908 binary questions (four per original question). We test 30 models from different families (e.g., OpenAI GPT, Llama, Qwen) across different model sizes (e.g., 8b, 70b, and 405b). We found this setup to be much more *challenging* for LLMs with the best performing model at only 61.5% accuracy and the worst at 21.4%, compared to a human performance of 92.6%.

Looking to understand why models perform drastically different on CULTURALBENCH-Easy and -Hard, we wondered if models can simply *guess* the most likely option under multiple-choice format found in the CULTURALBENCH-Easy setup. We designed an experiment that shows that models can get 40% accuracy (substantially above random chance of 25%) by simply choosing the option that has greatest embedding similar to the name of the culture (without seeing the question). This shows the potential limitation of assessing models' capabilities under the multiple-choice setting in CULTURALBENCH-Easy since they could rely on such heuristics without needing to demonstrate cultural understanding. In contrast, CULTURALBENCH-Hard can more *effectively* assess the cultural knowledge of models, because such heuristics cannot be easily applied to game evaluation.

Moreover, our evaluation on different question types shows that even the best models struggle with questions that have multiple correct answers, revealing a tendency to LLMs to over-converge on a single option. This is evident by a significant drop (-19.8%) in accuracy on questions with multiple correct answers, as compared with questions with a sole correct answer. Through our analysis of questions relating to various sub-continents in CULTURALBENCH-hard, we find that models perform well on questions relating to regions (e.g., North America and South Asia) that are highly represented in web-source data (e.g., United States, as part of North America) and large-scale human annotation sources (e.g., India in South Asia). However, models underperform on questions relating to less well-represented regions such as Eastern Europe. This observation holds even for models developed by providers based outside of the United States (e.g. Alibaba Qwen and Mistral), which might possibly be attributed to the availability of the data used in various stages of training. Overall, OpenAI's GPT-4o outperforms other proprietary providers and open source model builders uniformly across all but one region (Oceania). With CULTURALBENCH and our analysis on various models, we provide an effective benchmark for testing the cultural knowledge of various LLMs, with the hopes of encouraging model developers to easily perform cultural evaluations in the journey to develop more culturally-sensitive LLMs.

## 2 RELATED WORK

| Benchmark | 1. Robustness | | | 2. Diversity | | 3. How Challenging? | |
| --- | --- | --- | --- | --- | --- | --- | --- |
| | # Annotators per Qn (↑) | Verified Qn Coverage (Verified #/Total #) (↑) | Data Filtering by Majority Votes | Topic Inclusion | # Topic (↑) | Source | Best Model Performance (↓) |
| Candle (Nguyen et al., 2022) | 3 | 0% (0/1.1M) | ✗ | Predefined set | 6 | Web | 81.4% (GPT-3) |
| CultureAtlas (Fung et al., 2024) | 5 | 0% (0/10K) | ✗ | Predefined set | 8 | Wiki + LLM | 93.1% (GPT-3.5) |
| Normad (Rao et al., 2024) | 2 | 18.5% (480/2.6K) | ✗ | Predefined set | 4 | Web + LLM | 87.6% (GPT-4) |
| Blend (Myung et al., 2024) | 5 | 0% (0/500) | ✗ | Discovery-based | 5 | Human + LLM | 85.5% (GPT-4) |
| CULTURALBENCH (Our Work) | 5 | 100% (1227/1227) | ✔ | Discovery-based | 17 | Human + LLM | 61.5% (GPT-4o) (Human: 92.6%) |

Table 1: Comparison of existing cultural benchmarks on three criteria. Relative to existing benchmarks, CULTURALBENCH is *robust*, *diverse* and *challenging*. Verified Qn Coverage refers to the human quality checks on the final collected questions on the benchmark, rather than intermediate steps of data collection. Best Model Performance refers to the average accuracy/F1 scores attained by best performing model on benchmark, with the model in parenthesis.

Multicultural knowledge evaluation of LLMs have been widely investigated through building extensive knowledge bases (Shi et al., 2024; Keleg & Magdy, 2023); using socio-cultural surveys like World Value Survey (Durmus et al., 2023a; Tao et al., 2023; Ramezani & Xu, 2023); and generating more training data (Li et al., 2024). Here, we select four representative benchmarks with comparable model evaluation results, highlighting their limitations and the gaps that our CULTURALBENCH aims to fill in Table 1.

**Insufficient Quality Verification** Existing cultural benchmarks usually conduct quality check during the intermediate steps on data collection such as the relevance of web-scraped knowledge (Fung et al., 2024), commonality of knowledge (Nguyen et al., 2022). Blend asked humans to directly curate answers and aggregating those inputs to form questions but did not verify the final questions by humans (Myung et al., 2024). Normad verified part of the rule-of-thumbs but with two

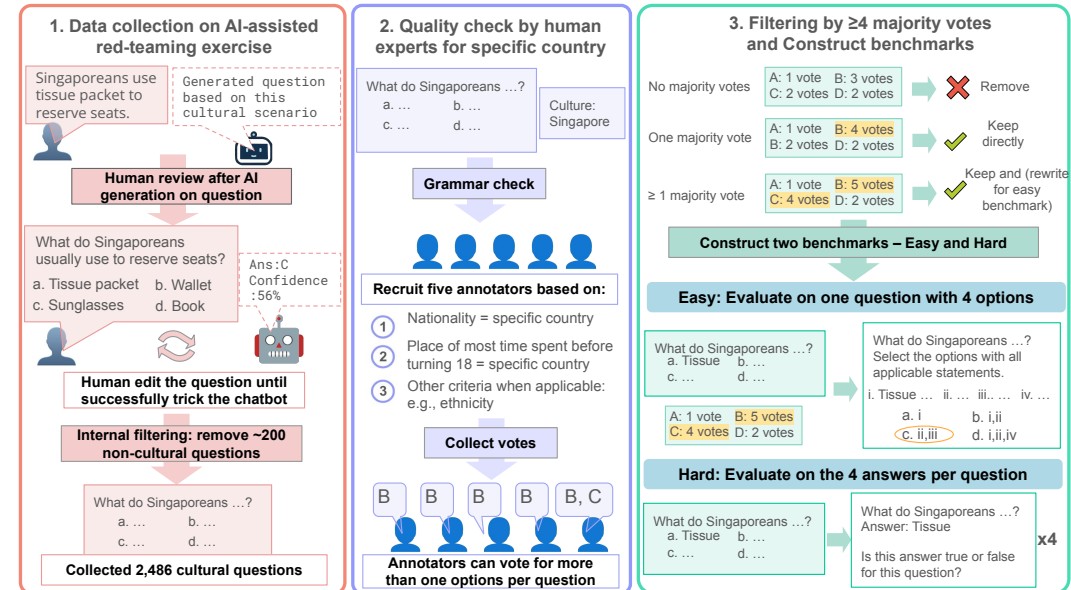

Figure 2: Step-by-Step details on Data Collection and Validation.

humans only (Rao et al., 2024). As cultural knowledge is not easily verifiable for correctness, it is essential to have reliable annotations on final set of questions (as given to LLMs) by having expert human verification on the full set of questions and then filtering out questions that does not reach consensus.

**Poor diversity of topics**  Many benchmarks have topics predefined prior to data collection, meaning that they are unlikely to fully capture the multi-faceted natured of cultural knowledge (Adilazuarda et al., 2024). Many prior works topics focus on narrow topics such as food (Nguyen et al., 2022), dating (Fung et al., 2024), social etiquette like dining (Palta & Rudinger, 2023; Dwivedi et al., 2023), visiting (Rao et al., 2024), and special elements in wider society like religions (Nguyen et al., 2022). To the best of our knowledge, only Blend uses a discovery-based approach to ask annotators to include all topics they believe to be relevant to culture(Myung et al., 2024), without restricting it to particular topics. CULTURALBENCH extends this discovery-based approach helping us to identify *diverse* topics outlined in Fig. 3.

**Over-reliance on Web Sources**  Existing benchmarks often rely on web sources directly such as web corpus (Nguyen et al., 2022), Wikipedia (Naous et al., 2023), and incorporated with LLMs' generation (Rao et al., 2024; Fung et al., 2024). These non-human written benchmarks may not be *challenging* since the scraped web sources may be used during models pretraining (Petroni et al., 2019) and LLM generations may inherit the potential cultural bias (Arora et al., 2022; Cao et al., 2023; Liu et al., 2024). Given the performances of best-performing models ranging from 81.4% to 93.1% in the existing benchmarks in Table 1, those benchmarks are likely not sufficiently *challenging* for modern frontier LLMs. Our proposed CULTURALBENCH is substantially more difficult with the best model (GPT-4o) only reaching 61.5% despite humans reaching 92.6%.

## 3 DATA COLLECTION PIPELINE

Our data collection pipeline consists of three steps, as illustrated in Fig. 2: (1) Data collection via AI-assisted red-teaming (2) Human quality check on full data (3) Filtering with majority vote. Such a multi-step process enables us to collect robust data for CULTURALBENCH.

## 3.1 STEP 1: DATA COLLECTION VIA INTERACTIVE AI-ASSISTED RED TEAMING

**Question Formulation.** Human annotators are instructed to brainstorm culturally relevant scenarios based on their personal experiences of their cultures (e.g., *Singaporeans use tissue packet to reserve seats*). A step-by-step guideline with detailed examples is provided to inspire them, as shown in Appendix H. The AI helper bot then transforms the scenario into a structured question with four options, which the annotators can review and edit afterward.

**Question Verification & Revision.** Human annotators can use the formulated question as basis to challenge the AI verifier in our interactive platform. The platform provide further assistance in revising the questions to make it more challenging by offering various revision strategies along with drafted examples (e.g., "Negate the Question"), as shown in Appendix H.

**Internal Filtering.** After collecting over 2,600 questions, the researchers carefully reviewed and removed those that are not relevant to any countries/regions (e.g., Bangladesh, Peru), resulting in a filtered set of 2,486 cultural questions.

## 3.2 STEP 2: HUMAN QUALITY CHECK

**Recruitment Criteria.** We collected questions at the country/regional level, pairing each question with a specific region. To ensure culturally attuned and thorough verification, we recruited five annotators for each region through the Prolific platform [1]. We set two main criteria to ensure that the recruited annotators have a deep understanding of the culture of the targeted country or region – (1) *Nationality* (2) *Primary residence before age 18*. For certain cultures (e.g. the United States, the United Kingdom), when the platform allowed more detailed selections and the collected question targeted specific groups in the country/region, we added detailed criteria such as *ethnicity* (e.g. African American, Native American), and *place of residence* (e.g., Wales).

**Multiple Selection Settings.** To better reflect the true representation of each cultural question, we allow annotators to select multiple answers on our questions with four options. As a result, some questions may have more than one majority-vote answer. This approach also helps test models' mode-seeking behavior, examining whether they rely solely on cultural stereotypes (i.e., modes) without considering broader cultural diversity.

## 3.3 STEP 3: FILTERING BY MAJORITY VOTE & CONSTRUCTING BENCHMARKS

**Majority Vote Criteria.** To build a robust benchmark that captures the accurate representation on cultural knowledge, we set the majority-vote threshold to be $>= 4$ out of 5 annotators. During human validation, we first filtered out questions without majority consensus, resulting in a final set of 1,227 questions. Subsequently, we further processed the remaining questions. To construct our CULTURALBENCH in two setups (CULTURALBENCH-Easy: Multiple-choice, CULTURALBENCH-Hard: True/False), we processed the questions differently depending on the numbers of majority votes they contain.

**(1) Single-Mode Questions (Only one majority vote).** For CULTURALBENCH-Easy, we directly keep the original question with four options. The gold label is the option with a majority vote (i.e., A, B, C or D). For CULTURALBENCH-Hard, we transform the question with four options into four binary questions. For instance, the question drafted (e.g., *"What do Singaporeans ...? A. Tissue ... D. ..."*) will form binary questions (e.g. *"Is this answer true or false for this question? Answer True or False only. Question: What do Singaporeans ...? Answer: Tissue."*).

**(2) Multi-Mode Questions (More than one majority votes).** For CULTURALBENCH-Easy, we reframe the question to allow multiple statements. The four drafted options (e.g., *"A. Tissue"*) become the four statements in questions (e.g., *"statement (i) Tissue"*). To ensure the models know the possibility of questions containing multiple correct labels, we add the instruction on question directly with (*"Select the options with all applicable statements"*). For CULTURALBENCH-Hard, we follow the same construction approach (transforming four options to four binary questions) as single-mode questions.

---

[1]https://www.prolific.com

## 4 CULTURALBENCH DESCRIPTION AND DISCOVERED TOPICS

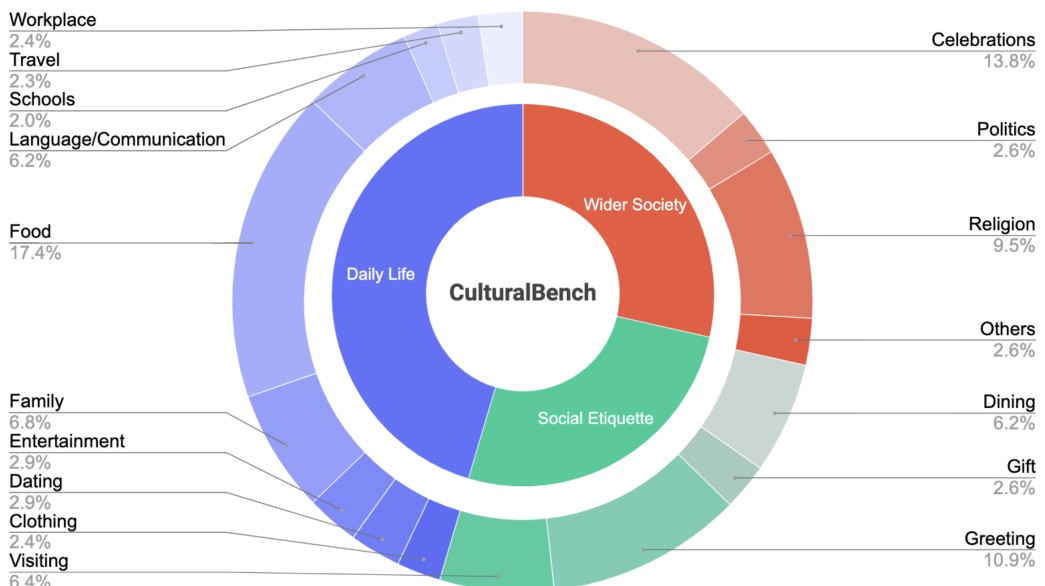

Figure 3: CULTURALBENCH covers 17 *diverse* cultural topics organized into three overarching categories.

### 4.1 DESCRIPTIVE STATISTICS ON CULTURALBENCH

Our benchmarks cover a **wide range** of global regions, spanning 45 countries and regions, including underrepresented regions such as Bangladesh, Zimbabwe, and Peru. A detailed breakdown of regional distribution can be found in Appendix B while example questions by topic are available in Appendix C.

**CULTURALBENCH-Easy.** It contains 1,227 multi-choice questions, each with four options. For instance, a question of *"What do Singaporeans usually use to reserve seats?"* with options of *"A. Tissue ... D. Book"* as shown in Fig. 1. The gold label is the correct option (A, B, C or D). For multi-mode questions (i.e., questions with more than one answer), we added an instruction of *"Selecting the option with all applicable statements"* to ensure that models consider all possible answers for fair evaluation. For instance, a question of *"What do Singaporeans...? Selecting the option ... statements. i) Tissue ... iv) Books"* with options of *"A. (i) ... D. (i), (ii), (iv)"*. The questions contain 17.2 words on average ($\sigma = 12.06$). Options at various positions have similar number of whitespace-separated words on average, specifically option A with 5.48 words ($\sigma = 4.24$), option B with 5.44 words ($\sigma = 4.27$), option C with 5.57 words ($\sigma = 4.24$), and option D with 5.57 words ($\sigma = 4.24$).

**CULTURALBENCH-Hard.** In this dataset, each question is transformed into four binary true/false questions, requiring models to evaluate each option separately. For example, the earlier multiple-choice example in CULTURALBENCH-Easy will transform into four binary questions such as *"Is this answer true or false for this question? Question: What do Singaporeans usually use to reserve seats? Answer: Tissue."*, as shown in Fig. 2. The gold label in this case is either True or False. This set contains 1,227 $\times 4 = 4,912$ True/False judgement questions. The questions contain 14.3 words ($\sigma = 5.27$) and the answers contain 5.72 words ($\sigma = 4.21$).

### 4.2 DIVERSE TOPICS DISCOVERED ACROSS CULTURES

Most existing cultural benchmarks have predefined topics to collect data on, typically on universal topics such as dining (Adilazuarda et al., 2024). However, this approach can overlook cultural elements unique to specific regions. To capture a broader spectrum of cultural topics, we adopted a discovery-based approach by encouraging human annotators to brainstorm cultural concepts from

their personal experiences. The detailed instruction for annotators can be found in Appendix H. CULTURALBENCH spans a *diverse* range of cultural elements with 17 topics under three categories (Daily life, Social Etiquette, and Wider Society), as shown in Fig. 3. Daily life relates to the everyday experiences of people e.g., Workplace. Social Etiquette means the acceptable norms in society e.g., Greeting. Wider Society included special elements for broader spectrum of cultural topics e.g., Celebrations. We classified questions into topics by prompting GPT-4o-mini. The classification prompt and the topic detailed definitions are in Appendix C.

To collect *diverse* data for each culture, we allow each annotator to create at most 3-7 questions, depending on the availability of annotators for each region. Notably, in curating CULTURALBENCH, we observed that people from different regions focused on distinct topics. For instance, annotators from Italy and Mexico provided more questions related to Food, with 15 out of 35 questions and 13 out of 49 questions respectively. In contrast, participants from South Africa and India focused more on Religion, contributing 19 out of 58 questions and 14 out of 46 questions respectively. Our discovery-based approach allow us to capture *diverse* cultural elements from people in different regions without being limited by a predefined set of topics.

## 5 EXPERIMENTS: EVALUATION OF LLMS ON CULTURALBENCH

We evaluate 30 current LLMs in a zero-shot setting on CULTURALBENCH in two setups: (1) CULTURALBENCH-Easy: Multiple choice; (2) CULTURALBENCH-Hard: True/False. We prompted the models to ensure they follow the output format to allow fair comparison. The detailed prompt is in Appendix D. To avoid exposing the correct answers to models for fair comparison, our annotation platform, which involves using OpenAI APIs did not allow the collected data to be used for further training.

**CULTURALBENCH-Easy.** We evaluate model performance by measuring accuracy, specifically whether the model correctly identifies the label for each multiple-choice question. A random baseline can achieve 25%.

**CULTURALBENCH-Hard.** We evaluate model performance based on the proportion of tasks in which the model can get all four options predicted correctly. For each task, an LLM has to make four binary judgements (True/False) from the transformation of four options in each multiple choice question. To demonstrate robust cultural knowledge, we believe the LLM has to accurately which option(s) are False as well as which option(s) are True. A random baseline can achieve $0.5^4 =$ 6.25%.

### 5.1 COMPARING LLMS ON TWO BENCHMARKS ACROSS MODEL FAMILY AND SIZE

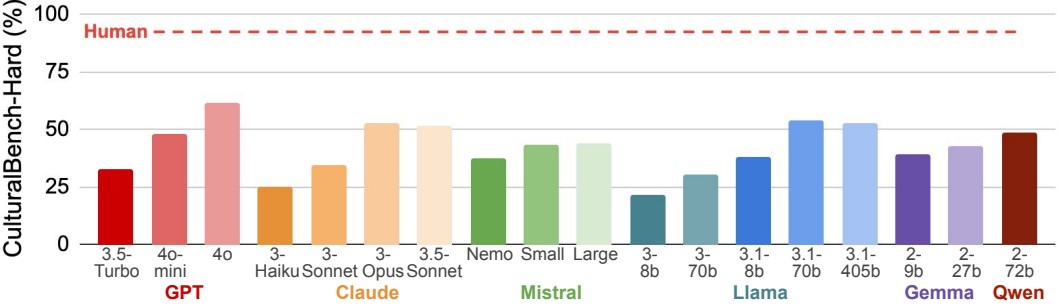

Figure 4: Models performance on CULTURALBENCH-Hard with random baseline at 6.25% and human performance at 92.6%.

We show the performance of 18 models across model families and sizes on CULTURALBENCH-Hard in Fig. 4. The corresponding Fig for CULTURALBENCH-Easy is in Appendix A.

**Models performance on CULTURALBENCH-Easy.** The best-performing model, GPT-4o, achieves 88.8% accuracy, slightly lagging behind human performance at 92.4%, as illustrated in Appendix A. Nonetheless, this benchmark remains an effective tool for assessing model capabilities, with the lowest score of 61.7% (Claude 3-Haiku) clearly highlighting the wide range of model performance.

**Models performance on CULTURALBENCH-Hard.** As shown in Fig. 4, this benchmark is significantly more ***challenging*** for current LLMs, with accuracy ranging from 21.4% for Llama3-8b to 61.5% for GPT-4o. These scores are considerably lower compared to the human baseline of 92.6%, highlighting the difficulty of the task even for the most advanced models.

**Models performance improves as model size increases.** In Fig. 4, we present the performance of models from six different families, such as GPT, Llama, and Qwen. Overall, the results demonstrate a trend of improved performance as model size increases. For example, within the Claude-3 family, the models show a clear progression in accuracy: Claude 3-Haiku achieves 25.3%, Claude 3-Sonnet reaches 34.5%, and Claude 3-Opus attains 52.9%. This pattern is consistent across most of the model families, indicating that larger models generally perform better on our CULTURALBENCH-Hard.

**Why do the two setups on CULTURALBENCH have such model performance difference?** We hypothesize that the models can guess for the most possible answer on CULTURALBENCH-Easy under the multiple-choice setting. We compute the embedding for the country name and separately for each option using OpenAI text-embedding-3-small. By using a simple heuristic of choosing the option with highest cosine similarity with the country name (e.g. Bangladesh), we attain 40.42% accuracy. This is intriguing as it is substantially above the random baseline (25%), without needing considering the question at all. We find that the cosine similarity difference between the correct option and the country name is significantly higher than the difference between options average and the country (0.166 vs. 0.145; Kruskal-Wallis $p$-value≤0.01). This shows the possibility of models guessing based on one (out of many possible) heuristics in multiple-choice setup without understanding (or even *knowing*) the question. This stresses the importance on using the binary (True/False) for each of the four options per question in CULTURALBENCH-Hard to accurately assess cultural knowledge of LLMs.

## 5.2 INVESTIGATING EFFECTS OF QUESTION TYPE AND TIME VERSION OF MODELS

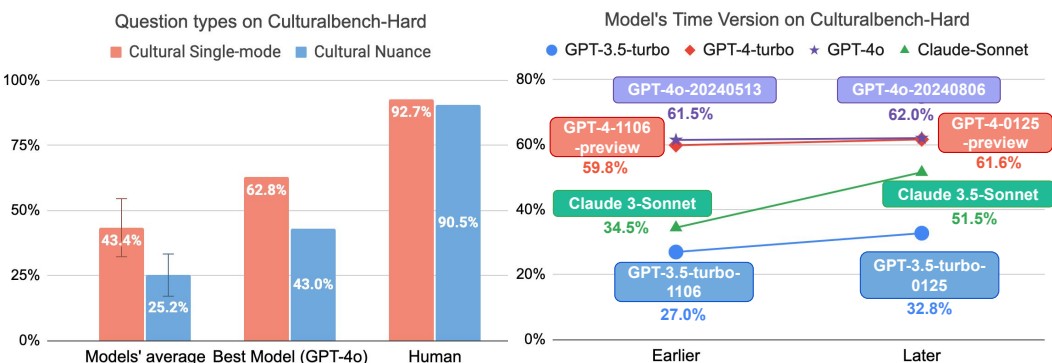

Figure 5: Analysis on question type (Left) and time version (Right). For question types, we demonstrate models struggle at answering questions with multi-modes (more than one correct answers). For time version, we show the improved performance of models across time.

**LLMs show distinct gaps between question types, unlike humans.** We evaluate the performance based on question types – (1) Single-mode (N=1141) and (2) Multi-mode (N=86). The first type refers to the questions with only one correct, majority-voted answer while the second type includes questions with multiple correct answers, as explained in Section 3.3. In Fig. 5 (Left), the average across all models shown in Fig. 4 is 43.4% on Single-mode questions and 25.2% on Multi-mode questions, revealing a significant gap of 18.2% between the two. Similarly, the best model (GPT-4o) exhibits a 19.8% performance difference between these question types. In contrast, human baselines show only a 2.2% difference, indicating that humans handle cultural diversity more effectively than models. This discrepancy suggests that models struggle to account for cultural nuances due to their mode-seeking tendencies, as discussed by (Tajwar et al., 2024).

**Models in the same series improve across time versions.** In Fig. 5 (Right), we evaluate four models (GPT-3.5-turbo, Claude Sonnet, GPT-4-turbo, and GPT-4o) across different available time versions (e.g., we evaluate GPT-3.5-turbo on 'GPT-3.5-turbo-1106' as earlier version and 'GPT-3.5-turbo-0125' as later version). Overall, all four models demonstrated an increasing trend in

performance across time. The largest improvement is shown in the Claude Sonnet model improves in performance from 34.5% (Claude-3 Sonnet) to 51.5% (Claude-3.5 Sonnet). By comparison, strongest model (GPT-4o) shows only a modest 0.5% increase between versions.

## 5.3 STUDYING DIFFERENT PROVIDERS' LLMS ON QUESTIONS FROM DIFFERENT REGIONS

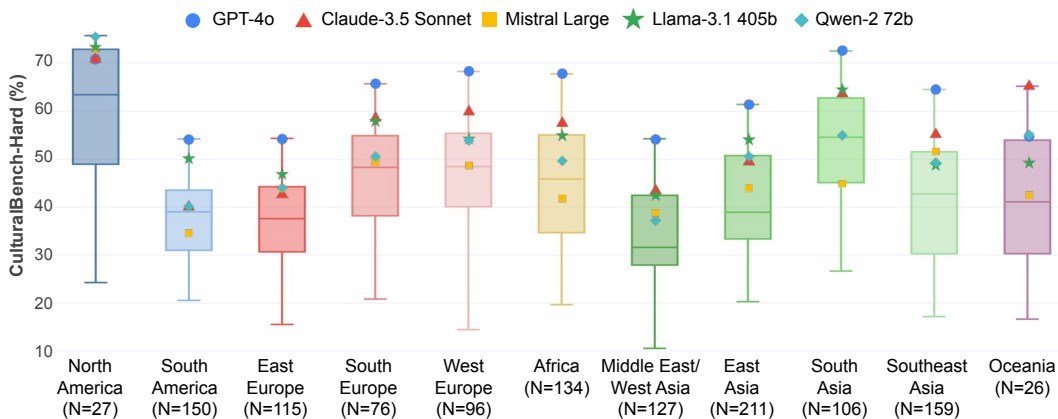

Figure 6: Models performance (18 models tested from Fig. 4) by different providers on different cultural groups. We further compare five representative models (GPT-4o, Claude Sonnet, Mistral Large, Llama-3.1 405b and Qwen-2 72b) from different model families.

We include detailed performance of models across different family and sizes (shown in Fig 4) to understand how well different models performance in questions relating to different geographic regions at a continent/sub-continent level.

**Overall, models perform better in questions relating to North America, South Asia, and West/South Europe.** From Fig. 4, it is evident that models achieve higher performance averages in regions like North America (58.0%), South Asia (52.3%), West Europe (47.1%) and South Europe (45.4%). We hypothesize that the higher performance in these regions can be attributed to several factors including their representation on web-data used for model training (Longpre et al., 2023) and the proportion of annotators recruited from these regions by LLM providers to curate post-training alignment data. For instance, many annotators are known to be recruited from India as they have good English ability and costs substantially less than their counterpart in the US (Lohchab & Roy, 2024).

**Models score lower in questions relating to South America, East Europe, and the Middle East.** Models exhibit lower performances on average in regions like South America (38.2%), East Europe (37.6%), and Middle East/West Asia (33.6%), compared to neighbouring regions such as North America (58.0%) and West Europe (47.1%). These disparities suggest insufficient representation of cultural knowledge from these regions in the training data.

**GPT-4o leads in most regions, followed by Llama-3.1 405b and Claude-3.5 Sonnet.** GPT-4o consistently ranks highest across most regions among all tested models. Llama-3.1 405b shows strength in regions where cultural knowledge is traditionally less represented, such as South America and East Europe, while Claude-3.5 Sonnet performs particularly well in other regions e.g., Oceania, West Europe, Africa, and Southeast Asia.

**Chinese Model Providers (Qwen-2-72b) and European Model Providers (Mistral Large) are not stronger in cultural knowledge relating in their region.** Despite claims of specialization in local languages, Qwen-2-72b and Mistral Large do not outperform other models in their respective regions in terms of cultural knowledge. For example, Qwen-2-72b scores 50.7% on East Asia, while GPT-4o achieves 61.4%. Similarly, Mistral Large underperforms in West Europe (48.9%) compared to GPT-4o (54.3%). These results suggest that local language proficiency alone is not sufficient for strong cultural competence.

## 6  LIMITATIONS AND FUTURE WORK

While CULTURALBENCH has several advantages over existing cultural benchmarks, we would also like to clarify some of its current limitations as well as ways to address them in the future.

**Multilingual vs. Multicultural.** We develop an English-only benchmark as the initial step in evaluating models' cultural knowledge. This approach facilitates fair comparisons of cultural understanding across different regions. For instance, in underrepresented regions such as Bangladesh, the availability of training data in local languages is often limited. As a result, models lacking sufficient exposure to these languages may struggle to comprehend questions phrased in them (Yong et al., 2023). By employing an English-only benchmark, we can assess models' cultural knowledge regarding these underrepresented areas without considering their (lack of) proficiency in low-resource languages. Additionally, prior research on multilingual models' emotional understanding (Havaldar et al., 2023) and reasoning skills (Liu et al., 2023) indicates that a model's multilingual capabilities may not necessarily correlate with its multicultural competencies. Notably, our discovery-based benchmark includes language elements on some questions, particularly in the Language and Communication topic with 6.2%. For example, we included questions like: *"What do Singaporeans usually say at the end? A. lah ..."*. As we await advances in developing stronger multilingual abilities in models for low-resource languages, our goal is to establish a robust, diverse, and challenging benchmark to track our progress toward addressing the uneven representation of cultural knowledge.

**Small sample of human verifiers on subjective cultural knowledge.** Due to the limitations of crowd-sourcing platforms like Prolific, the number of available annotators from underrepresented regions, such as Bangladesh, is quite small (fewer than 30 active human annotators). As a result, we were able to recruit only five annotators for consistency verification. To enhance the robustness of our dataset, we allow human verifiers to select multiple labels for each question, ensuring that all possible answers are captured. Additionally, we establish a strict majority-vote threshold (majority votes ≥ 4 out of 5). During the annotation process, we also provide two extra options: *"I don't have knowledge"* and *"This question is unanswerable"* – to enable annotators to indicate when they cannot provide a response.

**Further fine-grained culture classification.** We noticed that the country/region classification adopted by our CULTURALBENCH may not capture the cultural diversity within each region. However, the data annotation platform we accessed does not have a further fine-grain classification when recruiting human annotators for most of the regions except for the United States and the United Kingdom. To capture the diversity on these two countries, we revisited the data that have been filtered by having not enough majority votes and with mostly responses of *"I don't have knowledge"*. For example, questions asking for the Welsh custom in the United Kingdom may not be answerable for people living in England. Then, we conducted a second round of human quality check by assigning those questions for the specific groups of human annotators (e.g., people living in Wales in the United Kingdom), as explained in Section 3.2. We hope to see more data annotation tools for different local cultures to facilitate more fine-grained cultural data collection.

**Strong instruction prompts and strict evaluation criteria on models' outputs.** We evaluated models on zero-shot setting with the prompts. However, for some models such as Claude-3 Haiku, they need more instructions to have the right formatting. Therefore, we have added one-line instruction for all models to ensure they outputting the answer in the correct format on our evaluation, as described in Appendix D. However, with the strong instruction prompt, sometimes they still refuse to answer the questions e.g., *"This question ..."* rather than outputting the four options (i.e., *A, B, C, or D*) on CULTURALBENCH-Easy or the binary labels (i.e., *True/False*) on CULTURALBENCH-Hard. To ensure the fair evaluation, we set the output token to be 2. The model is treated as answering correctly when its output contains the correct labels only.

## 7  CONCLUSION

We present CULTURALBENCH in two setups: CULTURALBENCH-Easy and CULTURALBENCH-Hard. By establishing a robust, diverse, and challenging benchmark to track our progress in cultural knowledge, we hope it can motivate LLM providers to develop models that can be helpful to users across more geographical regions.

## ETHICS STATEMENT

Our data collection has been reviewed by university's IRB board to ensure it has no harm on human annotators. We pay annotators according to our vendor (Prolific)'s guidance, which is higher than the local wage requirement. Our annotation guidance has specifically asked annotators to not include their personal identifiable information when giving their responses. Before human verification, our internal team has reviewed the collected data to ensure there is no harmful or unsafe context such as sexual or violence content.

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

## A  RESULTS

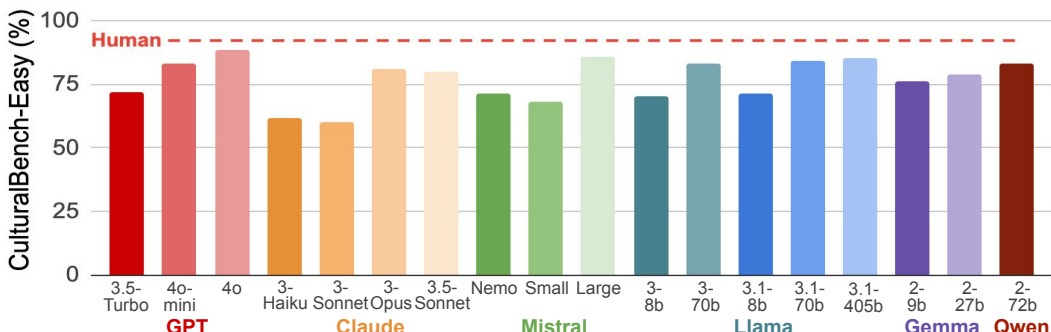

Figure 7: Models performance on CULTURALBENCH-Easy with random baseline at 25% and human performance at 92.4%

## B CULTURALBENCH STATISTICS

| Country | Counts |
|---|---|
| **North America** ($N = 27$) | |
| Canada | 7 |
| United States | 20 |
| **South America** ($N = 150$) | |
| Argentina | 35 |
| Brazil | 25 |
| Chile | 22 |
| Mexico | 49 |
| Peru | 19 |
| **East Europe** ($N = 115$) | |
| Czech Republic | 25 |
| Poland | 24 |
| Romania | 15 |
| Ukraine | 21 |
| Russia | 30 |
| **South Europe** ($N = 76$) | |
| Spain | 40 |
| Italy | 36 |
| **West Europe** ($N = 96$) | |
| France | 14 |
| Germany | 32 |
| Netherlands | 11 |
| United Kingdom | 25 |
| **Africa** ($N = 134$) | |
| Egypt | 20 |
| Morocco | 17 |
| Nigeria | 22 |
| South Africa | 58 |
| Zimbabwe | 17 |

| Country | Counts |
|---|---|
| **Middle East/West Asia** ($N = 127$) | |
| Iran | 37 |
| Israel | 13 |
| Lebanon | 22 |
| Saudi Arabia | 17 |
| Turkey | 38 |
| **South Asia** ($N = 106$) | |
| Bangladesh | 25 |
| India | 46 |
| Nepal | 21 |
| Pakistan | 14 |
| **Southeast Asia** ($N = 159$) | |
| Indonesia | 26 |
| Malaysia | 11 |
| Philippines | 45 |
| Singapore | 23 |
| Thailand | 27 |
| Vietnam | 27 |
| **East Asia** ($N = 211$) | |
| China | 59 |
| Hong Kong | 36 |
| Japan | 53 |
| South Korea | 41 |
| Taiwan | 22 |
| **Oceania** ($N = 26$) | |
| Australia | 15 |
| New Zealand | 11 |

Table 2: Country distribution of 45 countries in CULTURALBENCH

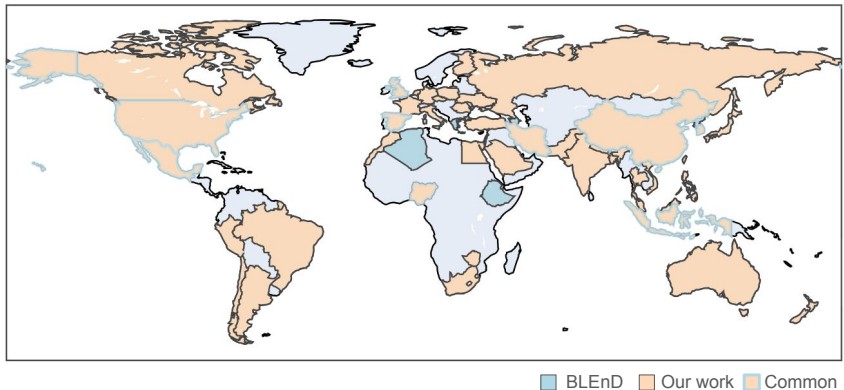

Figure 8: 45 Countries covered by CULTURALBENCH and 16 countries covered by Blend.

## C DEFINITION AND EXAMPLES OF CULTURAL TOPICS

- Daily life: universal concepts among cultures
    1. Clothing: Fashion, Jewelry trend;
    2. Food: cuisine, drinks;
    3. Entertainment e.g. game, movie, music, sports;
    4. Language/Communication e.g. linguistic, languages, general ways of communication that are not related to greeting people;
    5. Schools e.g. school setting, students, education;
    6. Workplace e.g. business setting, employee, employer;
    7. Travel/transport: ways of transport;
    8. Dating/marriage: committed relationship, romantic relationship;
    9. Family: children, parents, extended family relationship;

- Social Etiquette
    1. Greeting: what ways of greeting, gesture, how to greet people e.g. opposite sex, elderly;
    2. Dining: social rules related to dining e.g. which hands, how to pay;
    3. Gift: social rules about gift sending and receiving;
    4. Visiting and punctuality: rules about the concept of time, attending meetings, visiting friends;

- Wider society: specific concepts for certain cultures
    1. Celebrations e.g., celebrations, festival, holidays;
    2. Politics e.g. laws, rules by organization;
    3. Religion e.g. religious rituals, religious holidays;
    4. Others: any other topics or something in general.

| Topic | Example of question |
|---|---|
| **Daily Life** | |
| Clothing | What type of attire do Bengali males traditionally wear on Eid day?
A. Kurta Pajama
B. Suit and Tie
C. T-shirt and Jeans
D. Player's team apparel and gear |
| Dating | At what age do people commonly get married in German culture?
A. In their mid-30s
B. Right after high school
C. In their early 20s
D. Teenage years. |
| Entertainment | What is a popular activity among South Africans to relax during both summer and winter?
A. Going for a braai
B. Hosting a backyard barbecue
C. Heading to a beach bonfire
D. Camping |
| Family | What is not a common way for people to show respect to their grandparents in Moroccan culture?
A. Kiss their grandparents' hands
B. Kiss their grandparents on the forehead
C. Bow to their grandparents
D. Hug their grandparents |
| Food | In Indian culture, what is a common way of eating meals among locals?
Read the following statements and
select the option that include all the appropriate statements for this question.
(i) Using chopsticks
(ii) Eating with a fork and knife
(iii) Eating using hands/fingers
(iv) Eating with a spoon only.
A. ii, iv
B. iii, iv
C. i, iv
D. i, iii, iv |
| Language/
Communication | What do Singaporeans often append at the end of their sentences as a linguistic quirk
unique to their culture to show a form of exasperation?
A. "yah"
B. "lah"
C. "eh"
D. "yo" |
| Travel | In Indonesian culture, how do people mostly spend their time on the road?
A. Bicycles
B. Motorcycles
C. Cars
D. Scooters |
| Workplace | In Taiwanese culture, what is the most common question people will ask about your work?
A. Do you like your job?
B. How much do you make?
C. What are your job benefits?
D. Can you work remotely? |

Table 3: Examples of CULTURALBENCH for each topic.

| Topic | Example of question |
|---|---|
| **Social Etiquette** | |
| Dining | In Hong Kong culture, how do you signal to the server that you would like more water added to your tea while dining in local Chinese restaurants?
A. Remove the lid of the teapot
B. raise the cup and show to the server
C. Leave the teapot empty on the table
D. Point at the teapot |
| Greeting | In Yoruba culture of Nigera, how do young people traditionally greet elders?
A. Shake hands while maintaining eye contact.
B. Prostrate as a male and go on your knees as a female.
C. Bow slightly without making any physical contact.
D. Give a brief nod and smile. |
| Gift | In Russian culture, there are many gifts which are avoided because they represent bad omens or bad luck. Which of the following items is a very bad gift for your Russian friend?
A. Even numbers of flowers less than a dozen
B. Artwork
C. Chocolate chip cookies
D. Baseball caps |
| Visiting | In Peru, what is the cultural expectation regarding arrival time for a well planned birthday celebration event?
A. Arriving just in time
B. Arriving 15-30 minutes late
C. Arriving an hour early
D. Arriving 5-10 minutes early |
| **Wider Society** | |
| Celebrations | What do Vietnamese grandparents usually gift their grandchildren for the traditional Lunar New Year celebration?
A. Cards with best wishes
B. Traditional Foods and Snacks
C. Monetary gifts
D. Educational Materials |
| Religion | In Pakistani culture, what is the custom for Muslims regarding prayers on a specific day of the week?
A. Praying at mosque on Sunday
B. Offering Friday prayer
C. Praying before lunch time
D. Meditating on Friday morning. |
| Politics | In South Korea, only men are required to join the military.
What are the alternative civic duties that can be performed instead of military service?
A. Enrollment in educational programs for two years.
B. Volunteering in community services for a year.
C. Taking internship.
D. None of the options |
| Others | How many seasons are traditionally recognized in Bangladeshi culture?
A. 6 seasons
B. 4 seasons
C. 2 seasons
D. 5 seasons |

Table 4: (conc.) Examples of CULTURALBENCH for each topic.

# D ZERO-SHOT EVALUATION PROMPTS

**CULTURALBENCH-Easy**

Our evaluation is to ask the model in multiple choice setting. The zero-shot prompt is as follow to ensure the model only output one label (A, B, C or D).

To answer the following multiple-choice question, you should choose one option only among A,B,C,D. Instruction: You must select one option among A,B,C,D. Do not output any other things.

Question: <Question>

A. <Option A>

B. <Option B>

C. <Option C>

D. <Option D>

For multi-mode question, we included the instruction *"Select the options with all applicable statements"* to ensure models considering all statements provided.

**CULTURALBENCH-Hard**

Our evaluation is to ask the model in binary setting (True/False). Our prompt is as follow to ensure the model only output one label (True/False).

Question: <Question>

Answer: <Answer>

Is this answer true or false for this question? You must choose either True or False.'

# E  EVALUATION RESULTS ON CULTURALBENCH-EASY

| Models | North America | South America | East Europe | South Europe | West Europe | Africa | Middle East/ West Asia | South Asia | Southeast Asia | East Asia | Oceania |
|---|---|---|---|---|---|---|---|---|---|---|---|
| gpt-3-5-turbo-1106 | 80.72 | 71.34 | 72.64 | 83.75 | 78.61 | 81.22 | 54.71 | 80.68 | 67.5 | 63.7 | 86.36 |
| gpt35turbo | 87.85 | 72.21 | 76.86 | 80.97 | 74.61 | 80.74 | 59.19 | 83.82 | 68.39 | 62.04 | 78.48 |
| gpt4omini | 90.35 | 78.54 | 89.08 | 89.16 | 87.31 | 87.24 | 71.4 | 90.46 | 81.2 | 79.14 | 87.57 |
| gpt4o | 100.0 | 87.82 | 91.93 | 91.66 | 97.22 | 91.81 | 80.78 | 90.13 | 86.07 | 86.44 | 92.12 |
| gpt-4o-2024-08-06 | 100.0 | 85.61 | 92.32 | 93.06 | 97.0 | 92.42 | 86.76 | 90.22 | 86.28 | 88.5 | 92.12 |
| gpt-4-0125-preview | 100.0 | 86.9 | 90.96 | 90.42 | 94.44 | 87.97 | 82.34 | 94.11 | 85.46 | 84.7 | 87.57 |
| gpt-4-1106-preview | 97.5 | 87.22 | 92.28 | 91.8 | 90.65 | 85.59 | 82.5 | 95.65 | 84.95 | 85.6 | 96.66 |
| haiku | 80.35 | 58.8 | 64.91 | 67.91 | 55.78 | 72.39 | 58.07 | 68.76 | 57.09 | 56.52 | 76.37 |
| sonnet3 | 68.22 | 59.53 | 64.76 | 64.16 | 56.56 | 60.68 | 58.91 | 67.34 | 62.31 | 54.51 | 57.28 |
| opus | 100.0 | 74.56 | 81.16 | 84.86 | 83.96 | 83.13 | 73.58 | 88.64 | 80.1 | 79.47 | 92.12 |
| sonnet35 | 95.0 | 76.45 | 76.7 | 82.36 | 82.74 | 80.64 | 77.01 | 85.68 | 77.95 | 80.98 | 88.79 |
| mistralnemo | 80.72 | 69.46 | 78.49 | 74.86 | 72.83 | 77.66 | 62.26 | 79.63 | 71.68 | 64.82 | 80.91 |
| mistralsmall | 73.22 | 64.58 | 73.46 | 72.36 | 70.05 | 77.91 | 56.36 | 71.89 | 66.32 | 62.75 | 75.16 |
| mistral-large-2402 | 63.57 | 56.19 | 64.19 | 61.94 | 55.78 | 62.71 | 41.14 | 63.52 | 60.79 | 50.95 | 42.72 |
| mistrallarge | 95.0 | 82.87 | 88.18 | 93.2 | 86.53 | 88.9 | 79.13 | 88.68 | 82.45 | 84.77 | 92.12 |
| llama3-8b | 80.35 | 66.7 | 73.79 | 74.86 | 66.83 | 78.02 | 57.75 | 78.06 | 76.04 | 60.07 | 71.82 |
| llama3-70b | 97.5 | 78.54 | 87.9 | 87.64 | 85.74 | 85.37 | 75.94 | 86.34 | 85.66 | 78.65 | 79.7 |
| llama3-1-8b | 85.35 | 69.87 | 73.92 | 70.84 | 70.4 | 77.77 | 58.46 | 84.07 | 74.86 | 62.38 | 69.7 |
| llama3-1-70b | 97.5 | 76.29 | 88.43 | 87.78 | 86.53 | 83.02 | 76.98 | 91.65 | 87.39 | 80.15 | 92.12 |
| llama3-1-405b | 100.0 | 80.23 | 85.31 | 86.11 | 88.31 | 88.8 | 80.97 | 89.56 | 84.36 | 88.24 | 87.57 |
| gemma2-9b | 87.85 | 70.11 | 82.81 | 86.53 | 78.96 | 81.18 | 68.54 | 81.08 | 78.52 | 67.92 | 76.37 |
| gemma2-27b | 87.85 | 72.66 | 84.63 | 85.0 | 84.31 | 85.03 | 69.07 | 90.59 | 80.03 | 71.28 | 76.37 |
| mistral-7b-v1 | 61.43 | 53.36 | 57.2 | 56.25 | 58.92 | 68.35 | 49.33 | 63.76 | 60.71 | 55.0 | 56.06 |
| mistral-7b-v2 | 73.22 | 51.56 | 55.45 | 57.64 | 54.14 | 66.8 | 43.44 | 58.6 | 54.08 | 47.24 | 60.61 |
| mixtral-8x22B | 80.72 | 69.83 | 78.36 | 80.0 | 75.4 | 81.04 | 64.61 | 83.49 | 74.98 | 68.13 | 76.37 |
| qwen1-5-72b-chat | 97.5 | 74.54 | 85.48 | 90.56 | 82.18 | 83.03 | 71.01 | 88.01 | 81.32 | 76.98 | 80.91 |
| qwen2-72b | 97.5 | 81.45 | 86.94 | 86.39 | 89.87 | 85.78 | 69.86 | 87.59 | 78.14 | 83.03 | 95.46 |
| random | 24.28 | 29.85 | 22.76 | 26.39 | 32.39 | 28.26 | 25.32 | 20.8 | 25.59 | 22.9 | 25.76 |
| human | 91.65 | 92.27 | 93.6 | 92.92 | 92.31 | 91.65 | 92.53 | 94.48 | 93.47 | 92.08 | 90.06 |

Table 5: Accuracy (%) for 30 tested models on CULTURALBENCH-Easy at continent level.

## F  EVALUATION RESULTS ON CULTURALBENCH-HARD

| Models | North America | South America | East Europe | South Europe | West Europe | Africa | Middle East/ West Asia | South Asia | Southeast Asia | East Asia | Oceania |
|---|---|---|---|---|---|---|---|---|---|---|---|
| gpt-3-5-turbo-1106 | 36.78 | 26.45 | 20.21 | 29.58 | 37.4 | 28.37 | 17.61 | 36.74 | 25.33 | 23.38 | 36.97 |
| gpt35turbo | 41.78 | 36.27 | 30.65 | 34.03 | 43.53 | 34.22 | 21.0 | 46.51 | 30.28 | 28.76 | 36.97 |
| gpt4omini | 63.57 | 45.54 | 37.04 | 50.0 | 55.56 | 48.76 | 37.87 | 60.13 | 50.08 | 47.44 | 44.84 |
| gpt4o | 70.72 | 54.19 | 54.3 | 65.69 | 68.25 | 67.76 | 54.22 | 72.5 | 64.49 | 61.38 | 54.84 |
| gpt-4o-2024-08-06 | 75.72 | 55.76 | 52.73 | 64.03 | 69.81 | 65.2 | 59.05 | 68.42 | 61.39 | 62.49 | 60.61 |
| gpt-4-0125-preview | 75.72 | 57.42 | 56.57 | 60.14 | 68.03 | 67.56 | 54.94 | 66.7 | 59.99 | 65.22 | 59.39 |
| gpt-4-1106-preview | 77.85 | 52.82 | 53.9 | 61.25 | 60.12 | 66.87 | 53.23 | 73.5 | 59.56 | 61.55 | 56.06 |
| haiku | 43.93 | 30.17 | 28.36 | 20.84 | 33.96 | 27.64 | 19.42 | 33.0 | 17.2 | 20.32 | 27.88 |
| sonnet3 | 48.93 | 31.01 | 33.1 | 52.08 | 40.09 | 34.68 | 26.3 | 40.47 | 26.67 | 32.63 | 30.3 |
| opus | 75.35 | 43.52 | 47.59 | 54.86 | 52.78 | 58.94 | 42.45 | 64.37 | 58.28 | 52.95 | 53.94 |
| sonnet35 | 70.72 | 40.32 | 42.9 | 58.61 | 59.92 | 57.56 | 43.69 | 63.64 | 55.25 | 49.63 | 65.16 |
| mistralnemo | 48.93 | 30.09 | 37.84 | 46.94 | 45.66 | 39.86 | 29.76 | 45.78 | 39.63 | 34.06 | 34.84 |
| mistralsmall | 63.22 | 43.4 | 39.54 | 46.25 | 48.0 | 53.2 | 30.37 | 60.24 | 44.62 | 33.36 | 36.97 |
| mistral-large-2402 | 63.22 | 39.53 | 50.15 | 52.22 | 42.3 | 52.84 | 40.46 | 62.88 | 58.72 | 44.79 | 48.48 |
| mistrallarge | 72.85 | 34.93 | 44.11 | 49.58 | 48.88 | 42.08 | 39.13 | 45.09 | 51.78 | 44.22 | 42.72 |
| llama3-8b | 24.28 | 20.56 | 15.56 | 26.25 | 14.48 | 19.67 | 10.57 | 26.67 | 25.43 | 26.05 | 19.09 |
| llama3-70b | 27.14 | 26.2 | 24.09 | 23.75 | 34.52 | 30.02 | 27.92 | 38.54 | 30.01 | 33.73 | 41.52 |
| llama3-1-8b | 51.43 | 33.81 | 37.35 | 44.58 | 34.3 | 38.75 | 28.74 | 47.43 | 40.85 | 37.02 | 40.61 |
| llama3-1-70b | 75.72 | 46.18 | 45.53 | 56.25 | 63.69 | 60.15 | 50.04 | 62.72 | 51.5 | 57.19 | 53.94 |
| llama3-1-405b | 73.22 | 50.27 | 47.05 | 57.92 | 54.34 | 55.01 | 42.71 | 64.46 | 48.91 | 54.15 | 49.39 |
| gemma2-9b | 51.43 | 37.71 | 30.69 | 40.84 | 40.08 | 42.89 | 29.62 | 54.04 | 39.62 | 36.24 | 16.66 |
| gemma2-27b | 65.72 | 42.25 | 37.01 | 38.2 | 55.35 | 50.65 | 32.82 | 60.94 | 36.51 | 40.82 | 30.3 |
| mistral-7b-v1 | 31.78 | 26.2 | 18.53 | 22.08 | 22.39 | 22.72 | 11.43 | 21.71 | 15.67 | 16.68 | 15.76 |
| mistral-7b-v2 | 41.78 | 28.49 | 26.45 | 34.03 | 36.96 | 38.46 | 27.84 | 47.3 | 38.88 | 33.22 | 33.64 |
| mixtral-8x22B | 68.22 | 37.08 | 39.08 | 44.3 | 53.01 | 50.73 | 37.73 | 66.94 | 43.24 | 38.46 | 38.18 |
| qwen1-5-72b-chat | 75.35 | 45.88 | 40.48 | 45.84 | 44.22 | 49.83 | 29.83 | 65.22 | 44.87 | 41.15 | 41.52 |
| qwen2-72b | 75.35 | 40.44 | 44.23 | 50.7 | 54.0 | 49.83 | 37.52 | 55.04 | 49.48 | 50.72 | 55.16 |
| random | 21.78 | 5.7 | 9.98 | 5.28 | 3.34 | 4.95 | 8.01 | 8.07 | 5.2 | 5.92 | 3.34 |
| human | 94.0 | 91.26 | 92.58 | 94.26 | 92.84 | 94.0 | 91.48 | 94.29 | 92.2 | 92.57 | 93.29 |

Table 6: Accuracy (%) for 30 tested models on CULTURALBENCH-Hard at continent level.

# G ANNOTATORS' DEMOGRAPHICS IN PROLIFIC PLATFORM

| | North America | South America | East Europe | South Europe | West Europe | Africa | Middle East/ West Asia | South Asia | Southeast Asia | East Asia | Oceania |
|---|---|---|---|---|---|---|---|---|---|---|---|
| Gender (%) | | | | | | | | | | | |
| Female | 52.76 | 41.73 | 55.26 | 43.21 | 32.5 | 56.22 | 50 | 39.47 | 57.26 | 65.79 | 53.27 |
| Male | 46.62 | 58.26 | 44.74 | 56.79 | 67.5 | 42.49 | 48.95 | 59.87 | 42.74 | 32.89 | 46.73 |
| Prefer not to say | 0.61 | 0 | 0 | 0 | 0 | 1.29 | 1.05 | 0.66 | 0 | 1.32 | 0 |
| Age (%) | | | | | | | | | | | |
| ≤ 29 | 28.83 | 40.08 | 55.79 | 40.74 | 40.35 | 52.36 | 52.63 | 54.62 | 23.91 | 45.39 | 32.71 |
| 30-39 | 26.38 | 40.08 | 28.42 | 24.89 | 29.82 | 34.44 | 35.79 | 39.47 | 30.43 | 34.87 | 32.71 |
| 40-49 | 27.61 | 14.46 | 12.63 | 25.93 | 16.37 | 8.58 | 6.32 | 4.61 | 32.61 | 13.16 | 23.36 |
| 50-59 | 14.11 | 3.72 | 3.16 | 6.17 | 8.77 | 4.29 | 1.58 | 1.32 | 21.74 | 6.58 | 5.61 |
| 60-69 | 1.84 | 1.65 | 0 | 2.47 | 4.09 | 0.43 | 3.68 | 0 | 26.09 | 0 | 4.67 |
| 70-79 | 1.23 | 0 | 0 | 0 | 0.58 | 0 | 0 | 0 | 43.48 | 0 | 0.93 |
| Student status (%) | | | | | | | | | | | |
| No | 85.28 | 64.88 | 56.84 | 51.85 | 76.02 | 49.79 | 45.26 | 48.03 | 60.48 | 61.18 | 66.36 |
| Yes | 9.82 | 30.99 | 37.89 | 46.91 | 22.22 | 45.92 | 45.26 | 46.71 | 31.05 | 23.68 | 24.30 |

Table 7: Annotators demographic in Prolific for the whole dataset before filtering. We set two main recruitment criteria to ensure the recruited annotators have a deep understanding of culture of the targeted country or region: (1) Nationality. (2) Primary residence before age 18. See the detail in Section 3.2.

## H    AI-ASSISTED RED TEAMING SYSTEM

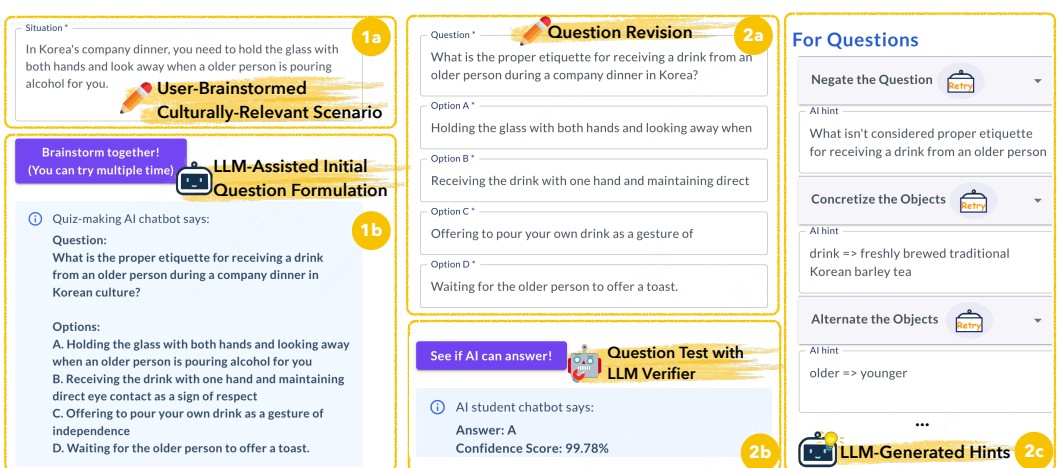

Figure 9: Interface for Step 1 (Data collection via Interactive Red teaming). **(1a)** Users brainstorm culturally relevant scenarios **(1b)** They convert scenarios to MCQs with LLM-powered Question Formulation **(2a)** Users revise MCQs and **(2b)** test MCQs based on the chosen option and its confidence score from LLM Verifier **(2c)** Users inspire by LLM-generated hints with strategies e.g., Negation, Synonym. (TODO: change order)

**Step 1: Data collection via interactive AI-assisted red teaming**

This system consists of two steps, as demonstrated in Fig. 9 – 1) Question Formulation 2) Question Verification and Revision 3) Feedback Collection. The first two steps involve a red-teaming exercise to formulate a challenging question step-by-step.

**Step 1a: Question Formulation** The goal is to facilitate users in brainstorming culturally relevant situations based on their personal experiences. A step-by-step guideline with detailed examples is provided to inspire them, as shown in Fig. 10. Users formulate a multiple-choice question (MCQ), which comprises one correct and culturally appropriate option.

**Step 1b: Question Verification & Revision** This step provides an interactive and iterative red-teaming platform that allows users to verify their culturally sensitive MCQs. The platform assists them in revising the question and the options to make it more challenging by providing descriptions of various common revision strategies with drafted examples (e.g., "Negate the Question"), as stated in Fig. 9 and Fig. 11.

## 1. What makes your culture so different from US mainstream culture?

It could be social behaviors, traditions, customs, or norms... Reflect on your personal expereience/habit in daily life or moments of cultural shock you encountered when exposed to US culture, or aspects you believe would surprise people from the US.

---

**Examples** +

- **social behaviors**
  Mediterranean culture: People tend to talk louder during meal in public.
  US mainstream culture: It is not common. Talking loud sounds they are arguing.

- **traditions**
  Mexican culture: They celebrate girl's 15th birthday as transition into womanhood.
  US mainstream culture: They do not treat 15th birthday as transition into womanhood.

- **customs**
  Japanese culture: People remove your shoes before entering someone's home to show respect.
  US mainstream culture: People often keep their shoes on indoors.

- **norms**
  Indian culture: Greeting others with a "Namaste" gesture - pressing palms together, and bowing.
  US mainstream culture: Handshakes or hugs are more commonly used as greetings.

---

Figure 10: Detailed Guidance on Step 1a (brainstorming culturally-relevant scenario) in our interactive red teaming system.

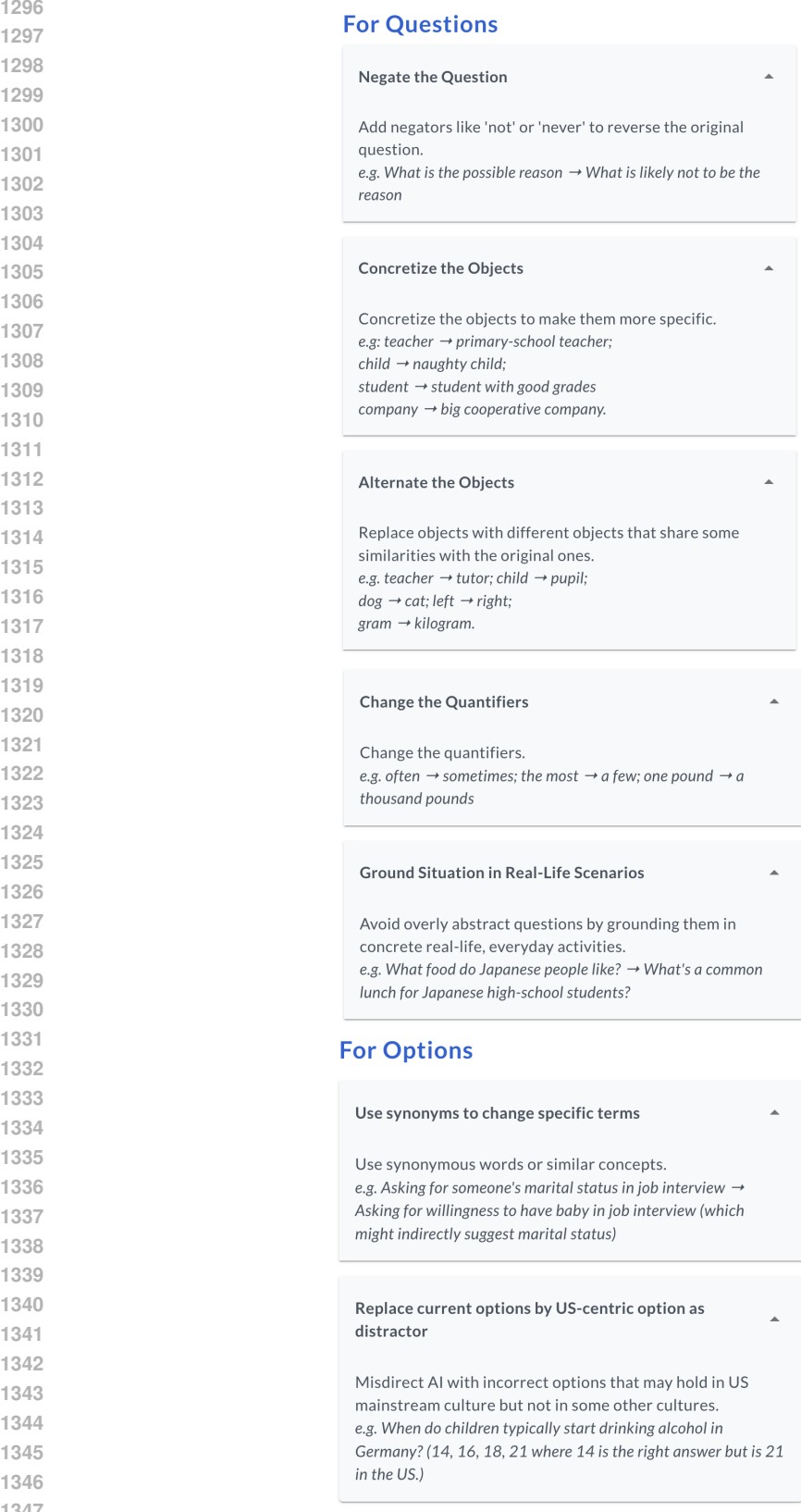

## For Questions

**Negate the Question** ▲

Add negators like 'not' or 'never' to reverse the original question.
*e.g. What is the possible reason → What is likely not to be the reason*

**Concretize the Objects** ▲

Concretize the objects to make them more specific.
*e.g: teacher → primary-school teacher;*
*child → naughty child;*
*student → student with good grades*
*company → big cooperative company.*

**Alternate the Objects** ▲

Replace objects with different objects that share some similarities with the original ones.
*e.g. teacher → tutor; child → pupil;*
*dog → cat; left → right;*
*gram → kilogram.*

**Change the Quantifiers** ▲

Change the quantifiers.
*e.g. often → sometimes; the most → a few; one pound → a thousand pounds*

**Ground Situation in Real-Life Scenarios** ▲

Avoid overly abstract questions by grounding them in concrete real-life, everyday activities.
*e.g. What food do Japanese people like? → What's a common lunch for Japanese high-school students?*

## For Options

**Use synonyms to change specific terms** ▲

Use synonymous words or similar concepts.
*e.g. Asking for someone's marital status in job interview → Asking for willingness to have baby in job interview (which might indirectly suggest marital status)*

**Replace current options by US-centric option as distractor** ▲

Misdirect AI with incorrect options that may hold in US mainstream culture but not in some other cultures.
*e.g. When do children typically start drinking alcohol in Germany? (14, 16, 18, 21 where 14 is the right answer but is 21 in the US.)*

Figure 11: Detailed Guidance on Step 1b (Question verification & revision) in our interactive red teaming system.

