# OpenReview forum: "CulturalBench: a Robust, Diverse and Challenging Benchmark on Measuring (the Lack of) Cultural Knowledge of LLMs"
_ICLR.cc/2025/Conference — Submitted to ICLR 2025_

### Official Review · Reviewer_aDzz · 2024-10-27

**Soundness:** 3
**Presentation:** 3
**Contribution:** 3
**Rating:** 5
**Confidence:** 4

**Summary:**

This paper proposes CulturalBench, an AI-assisted red-teaming collected dataset. The dataset aims to help improve the cultural understanding ability of LLMs, which is so meaningful. Besides, the datasets is collected by local people, covering lots of areas and topics. The experiments in the paper also improve the effectiveness of the dataset.

**Strengths:**

1. The paper proposes a pipeline for collect cultural data leveraging AI-assisted red-teaming.

2. The human collecting process is professional and considerable.

**Weaknesses:**

1. I am confusing about the experiment in the last part of sec.5.1. You supposed that models perform better on the easy version because models can guess the answer of some questions without seeing the questions. Your experiment is to compute the semantic similarity of the cultural name with each answer. I think it may only prove the cultural relativity of each pair. A better exploration may be to control the variables. For example, disturb the choices of each question to change their cultural relativity, and see the new results. For example, you could conduct this additional experiment of randomizing answer choices and re-evaluating model performance, to more conclusively demonstrate whether models are relying on cultural relativity heuristics.

2. Can CulturalBench helps to enhance models' cultural understanding? More experiments on prompting or fine-tuning may be more interesting. For example, you can fine-tune models using your data and conduct in-context learning and other prompt engineerings.

**Questions:**

1. According to sec.3.2, how to define the correct answers for a question with multiple answers considering cultural diversity?

---

> ### Author Response · Authors · 2024-11-23
> **Response to Reviewer aDzz – (1/3)**
>
> We thank the reviewer for your feedback and questions.
>
> > I am confusing about the experiment in the last part of sec.5.1. You supposed that models perform better on the easy version because models can guess the answer of some questions without seeing the questions. Your experiment is to compute the semantic similarity of the cultural name with each answer. I think it may only prove the cultural relativity of each pair. A better exploration may be to control the variables. For example, disturb the choices of each question to change their cultural relativity, and see the new results. For example, you could conduct this additional experiment of randomizing answer choices and re-evaluating model performance, to more conclusively demonstrate whether models are relying on cultural relativity heuristics.
>
> We apologized for the reviewer's confusion. The purpose of doing the experiment was to show if models can guess the correct answer to a question, **without even seeing the question**. We showed that it can somewhat do this (with 40% accuracy) by relying on the heuristic of guessing based on the semantic similarity between the country name and candidate options. This forms the motivation for us to construct CulturalBench-Hard, which disallows models to guess using such heuristics.
> We are not sure what the reviewer refers to as ‘only prove the cultural relativity of each pair’ because cultural relativity is very much a core aspect of cultural knowledge that we should measure. In the context of measuring cultural knowledge of models, it is reasonable to measure if  “cultural relativity of each pair” provides a shortcut for models.
>
> Given this example below:
>
> What's a common breakfast in China?
>
> A. Youtiao
>
> B. Doughnut
>
> C. Overnight oats
>
> D. Bacon and Eggs
>
> → correct option: A
>
> Because by ‘perturbing’ the correct option (i.e. A. Youtiao to become A. Ramen), the question is now unanswerable (since none of the options are correct).
>
> A. Ramen
>
> B. Doughnut
>
> C. Overnight oats
>
> D. Bacon and Eggs
>
> By ‘perturbing’ an incorrect option (i.e. B. Doughnut to become B. Soymilk), option B is now also correct alongside option A, which again makes the question unanswerable.
>
> A. Youtiao
>
> B. Soymilk
>
> C. Overnight oats
>
> D. Bacon and Eggs
>
> The above example shows it is hard to add perturbation to question options in an automatic manner as it requires humans (experts in the particular culture) to verify if the perturbation does not cause unintended consequences. It might be possible to perturb incorrect options to give easy negatives (e.g. B. Doughnut to become B. Bicycle, which might come from another question unrelated to food), but it might not be useful for our goal of building a challenging dataset as that perturbed option is an ‘easy negative’. If the reviewer have other ideas on how the answer choices can be automatically 'perturbed' (without needing additional human verification), we will be happy to hear more.

---

> ### Author Response · Authors · 2024-11-23
> **Response to Reviewer aDzz – (2/3)**
>
> > Can CulturalBench helps to enhance models' cultural understanding? More experiments on prompting or fine-tuning may be more interesting. For example, you can fine-tune models using your data and conduct in-context learning and other prompt engineerings.
>
> CulturalBench is a test dataset so if we fine-tune on the test data and then evaluate it again on the same dataset, it will show a high score on the dataset, without the score necessarily reflecting the general cultural knowledge of the trained model. **Our goal is to create a diverse and robust evaluation dataset, which enables LLM providers to measure the cultural knowledge of their models.** We believe that setting an effective metric will inspire subsequent efforts to collect further training data for culture - as HumanEval [1] did for coding, GSM8K [2] did for math and BFCL [3] did for function calling.
>
> **Prompting experiment.** We thank the reviewer for suggesting a prompting experiment on our CulturalBench. Following the reviewer’s suggestion, we tried different prompt variations on the best-performance model (GPT-4o) in our CulturalBench-Hard due to the limitation of time and budget. The prompts details are as follows:
>
> Default prompt:
> ```
> Question:{question}\nAnswer:{option}\nIs this answer true or false for this question?\nYou must choose either True or False.
> ```
>
> Inspired by Global Opinion paper [4], we tried three variations of their cross-national prompt using different perspectives.
>
> Cross-national prompt 1 (perspective: pretending to be the local):
> ```
> How would someone from {country} consider the following:\n Question:{question}\nAnswer:{option}\nIs this answer true or false for this question?\nYou must choose either True or False.'
> ```
> Cross-national prompt 2 (perspective: pretending to answer a local user):
> ```
> How should you respond if someone from {country} asks you the following:\n Question:{question}\nAnswer:{option}\nIs this answer true or false for this question?\nYou must choose either True or False.'
> ```
>
> Cross-national prompt 3 (perspective: pretending to be an expert of the culture):
> ```
> How would an expert on the culture of {country} consider the following:\n Question:{question}\nAnswer:{option}\nIs this answer true or false for this question?\nYou must choose either True or False.
> ```
>
> |        | **Default prompt** | **Cross-national prompt 1** | **Cross-national prompt 2** | **Cross-national prompt 3** |
> |--------|--------------------|-----------------------------|-----------------------------|-----------------------------|
> | GPT-4o | 61.5%              | 61.29%                      | 63.56%                      | 62.75%                      |
>
>
> The result by cross-national prompt 2 (pretending to answer a local user) has the best performance. However, the small rise of performance (2.06% from 63.56%-61.5%) indicates the prompting itself is not a significant factor in the model's performance on CulturalBench.
>
> We will add the above prompt exploration into Appendix for future reference. If the reviewer has other prompting methods that are useful, feel free to let us know.
>
> Reference:
>
> [1] https://arxiv.org/abs/2107.03374 Evaluating Large Language Models Trained on Code
>
> [2] https://arxiv.org/abs/2110.14168 Training Verifiers to Solve Math Word Problems
>
> [3] https://arxiv.org/abs/2305.15334 Gorilla: Large Language Model Connected with Massive APIs
>
> [4] https://arxiv.org/abs/2306.16388 Towards Measuring the Representation of Subjective Global Opinions in Language Models

---

> ### Author Response · Authors · 2024-11-23
> **Response to Reviewer aDzz – (3/3)**
>
> > According to sec.3.2, how to define the correct answers for a question with multiple answers considering cultural diversity?
>
> Having a multilabel setting **allows human annotators to indicate the co-exist cultural norms per question.**
>
> For instance, a question in our benchmark:
>
> Question: In Japanese culture, what is something that almost everyone knows about themselves?
>
> A. Their blood type
>
> B. Their social security number
>
> C. Their astrological sign
>
> D. Their credit score
>
> We asked five independent annotators to answer this question. They are allowed to select more than one option as correct answers. Here, Options A and C both attain majority votes (more than four out of five votes).
>
> To better understand our explanation, here some relevant information on cultural context in Japan:
> - Option A: blood type is a significant part of Japanese culture since Japanese people believe it can determine personality and even use it in job applications (started from 1970) [1].
> - Option C: For the younger generation, people were influenced by western culture and adapted some elements (e.g. astrological sign) to create works like the comic Saint Seiya (Seiya is “star” in translation) in 2000 [2], [3].
>
> This example shows the **diverse representation of Japanese culture (across age groups).** We will add this example for clarification.
>
> Reference:
>
> [1] https://www.bbc.com/news/magazine-20170787 Japan and blood types: Does it determine personality?
>
> [2] https://www.reddit.com/r/SaintSeiya/comments/1atbcyn/is_saint_seiya_still_relevant_in_japan/
>
> [3] https://www.rockandart.org/influence-anime-and-manga-western-pop-culture/ The Influence of Anime and Manga on Western Pop Culture

---

### Official Review · Reviewer_NUgQ · 2024-10-27

**Soundness:** 3
**Presentation:** 3
**Contribution:** 2
**Rating:** 5
**Confidence:** 4

**Summary:**

The paper describes work on developing a high-quality benchmark called CulturalBench for measuring cultural knowledge of LLMs. The benchmark contains a total of 1,227 high-quality and human-verified English prompts covering culture-based questions from 45 global regions and split into two versions of the benchmark: CulturalBench-Easy and CulturalBench-Hard. The paper introduces CulturalTeaming which is the process of constructing high-quality generation questions from cultures and validation based on majority voting. The paper highlights differences of CulturalBench from existing culture-evaluating benchmarks including quality checks for all 1,227 questions, discovery-based topics, and higher topic coverage. One particularly finding that stood out for me is that model providers' flagship models were not able to obtain higher performance in cultural knowledge relating to their region.

**Strengths:**

The paper is well-written and easy to read. I particularly appreciate the level of completeness of the paper and how it packages its contributions in terms of benchmarking cultural knowledge of LLMs. I find the CulturalTeaming process used to ensure quality of questions crafted by humans towards specific cultural practices as part of the benchmark to be one of the paper's strengths. The author/s highlight this difference compared to existing benchmarks where instances are not verified iteratively and strongly through majority voting. In terms of results, the paper has interesting findings such as learning that model providers from specific regions (ex. Qwen from Asia/China) seem not to meet the expectation that it will emerge as the best performing model for East Asian culture assessment.

**Weaknesses:**

Some of the issues I find from the paper are as follows:

1. **Small test instance count which has certain disadvantages.** 1,227 test instances may be a good initial overview of performance but not enough to tell something about an LLM’s true/reliable understanding of geographic-specific culture (ex. South Europe or Oceania) as it only contains 20-100 test instances (even less if you revert it back to its original question-type form). The quality control in developing the benchmark is an advantage (and I think this should be a common practice that other benchmarks can replicate) but I’m concerned about the practical usability of said benchmark as the trend in evaluating culture understanding is more favorable towards depth on a specific global region than a catch-all/one-size-fits-all evaluation.

2. **Selective results using commercial models.** The authors did perform in-depth analysis of the question type and versions of models in Section 5.2 but only for commercial models. This does not make sense to me as there is very little to know about these models due to their closed nature. It would make the paper stronger if the same analysis is done on more transparent/open-weight models used like Llama, Qwen and even models not included in the experiments but are made with culture in mind such as Aya models by Cohere.

3. **Lack of discussion on how to grow and develop the benchmark.** For culture knowledge benchmarks like this one, it might strengthen the paper if the authors provide sufficient and convincing discussion on how the community or any plans to grow and develop the benchmark, given its small number of test instances. This can be in the form of improving the framework in Figure 1. Without this, the benchmark will only cover a snapshot of cultures of different geographical areas which, again, makes its real-life usage questionable.

4. **Wrong accusation on poor diversity of previous benchmarks**. I oppose the author/s statement to label topic-specific benchmarks focusing specifically on food, social etiquette, etc as "narrow" and of  "poor diversity" in the related work section. As a reader, the tone of the paragraph comes as negative. While these benchmarks cover more specific aspects of culture, they have their own application as to why it is important to benchmark LLMs on different specific cultural aspects like food. They are considered semantic proxies of culture by a survey paper (Adilazuarda et al., 2024), which is a good starting point for understanding how LLMs capture cultures. This is not a negative connotation per se. Moreover, I would argue that it is better to have a benchmark that prioritizes depth on a specific aspect of culture but provides solid, empirical insights on that direction than a benchmark that tries to "catch-all" cultures but obtains very limited unrepresentative findings from small test instances.

**Questions:**

How is a binary question format classified as hard when there are more chances of a model picking either one correct from the two choices? I believe I’m missing something because this is not clearly explained in the paper. If a benchmark is hard/challenging, the first thing that comes to me is something similar to MMLU-Pro (https://arxiv.org/abs/2406.01574) where there are greater than 5 (max 10) choices.

See other questions raised in Weaknesses.

---

> ### Author Response · Authors · 2024-11-23
> **Response to Reviewer NUgQ – (1/3)**
>
> > Small test instance count which has certain disadvantages. 1,227 test instances may be a good initial overview of performance but not enough to tell something about an LLM’s true/reliable understanding of geographic-specific culture (ex. South Europe or Oceania) as it only contains 20-100 test instances (even less if you revert it back to its original question-type form). The quality control in developing the benchmark is an advantage (and I think this should be a common practice that other benchmarks can replicate)
>
> We thank the reviewer’s appreciation of our quality control in developing the benchmark.
>
> **Small but effective instances.** For evaluation datasets, having high quality is extremely important - more so than quantity. TruthfulQA [1] is trusted by over 1, 200 research work that cites it, including HF Open LLM v1 Leaderboard [2], despite being of the same size as CulturalBench. We are also inspired by the use of rigorous human validation in data curation for GPQA Diamond [3], which results in a small dataset with fewer than 200 questions. Despite its small size, many recent works such as OpenAI o1 [4] and Anthropic Claude 3.5 Sonnet (Oct 2024 version) [5] are choosing to benchmark their models using it due to its high quality over much larger but lower quality benchmarks such as MMLU [6] (with over 15k samples).
>
> We also want to clarify that the test instances per geographic-specific culture as the reviewer referred to is our “original question-type form” (see more detail in Figure 6). It means that South Europe has 76 questions for 76 different scenarios, and Oceania has 26 questions for 26 different scenarios. The total number of questions is 1,227 for both CulturalBench-Easy and CulturalBench-Hard. For CulturalBench-Hard, it has 1,227*4 binary test instances but we evaluate models at question level.
>
> > I’m concerned about the practical usability of said benchmark as the trend in evaluating culture understanding is more favorable towards depth on a specific global region than a catch-all/one-size-fits-all evaluation.
>
> > Wrong accusation on poor diversity of previous benchmarks. I oppose the author/s statement to label topic-specific benchmarks focusing specifically on food, social etiquette, etc as "narrow" and of "poor diversity" in the related work section. As a reader, the tone of the paragraph comes as negative. While these benchmarks cover more specific aspects of culture, they have their own application as to why it is important to benchmark LLMs on different specific cultural aspects like food. They are considered semantic proxies of culture by a survey paper (Adilazuarda et al., 2024), which is a good starting point for understanding how LLMs capture cultures. This is not a negative connotation per se.  Moreover, I would argue that it is better to have a benchmark that prioritizes depth on a specific aspect of culture but provides solid, empirical insights on that direction than a benchmark that tries to "catch-all" cultures but obtains very limited unrepresentative findings from small test instances.
>
> We appreciate the previous efforts on benchmarking and deep analysis for specific topics/regions. We will revisit the comparison paragraph to frame in a more constructive manner while showing our concern and necessity to have a high-quality and challenging cultural benchmark which contains diverse topics across various aspects of culture.
>
> As a community, we believe that cultural knowledge evaluations providing “a catch-all/one-size-fits-all evaluation” can complement other evaluations focusing “depth on a specific global region (or domain)”. The immediate problem that “a catch-all/one-size-fits-all evaluation” can alleviate is that while LLM builders such as OpenAI [4], Anthropic [5] and Meta [6] are evaluating their models on math, code and instruction following capabilities, they have not found a general cultural knowledge benchmark of sufficiently high quality and challenge to report their model performances on.
>
> In this sense, we are trying to provide a robust, diverse and challenging dataset for these LLM builders to start to consider evaluations on cultural knowledge. Without “a catch-all/one-size-fits-all evaluation” to start, we are worried that the initial barrier to immediately go in depth across all global regions/domains will be too high, which might make LLM builders disinclined to conduct such cultural knowledge evaluations altogether. Once LLM builders get started using a “a catch-all/one-size-fits-all evaluation”, they will be incentivised to go in “depth on a specific global region” (or domain), especially in regions that they serve their users in or domains that they are most interested in. Together, both types of evaluation can support LLM Builders to build models that have accurate representation of cultural knowledge.

---

> ### Author Response · Authors · 2024-11-23
> **Response to Reviewer NUgQ – (2/3)**
>
> > Selective results using commercial models. The authors did perform in-depth analysis of the question type and versions of models in Section 5.2 but only for commercial models. This does not make sense to me as there is very little to know about these models due to their closed nature. It would make the paper stronger if the same analysis is done on more transparent/open-weight models used like Llama, Qwen and even models not included in the experiments but are made with culture in mind such as Aya models by Cohere.
>
> We thank the reviewer for their suggestion on including more transparent/open-weight models (e.g. Llama,Qwen and Aya).
>
> **Question type analysis.** Following the reviewer’s suggestion, we also evaluate Aya models (8b and 32b) and report its results in the table below together with other open-weight models. We clarify that Figure 5 includes all 30 tested models (including open-weight models such as Llama-3, Llama-3.1, Gemma, Qwen). We initially focussed our discussion on GPT-4o as it was the best performing LLM on CulturalBench, but we will include further discussions on open-weight models going forward as we agree that more information is available for open-weight models. In particular, Aya 8B and 32B both seem to perform poorly, despite being made with ‘multilinguality’ (instead of culture in mind, as the reviewer suggests). This means that training models explicitly for multilingual capabilities does not guarantee its understanding of cultural knowledge.
>
> **Time version analysis.** While not included in Figure 5, we have shown a comparison between different time versions of Llama 3 (3 vs 3.1 for both 8B and 70B variants) in Figure 4. Llama 3 8B and 70B both demonstrate substantial gains on CulturalBench between time versions (3 to 3.1), suggesting that open-weights models are progressing rapidly in terms of cultural knowledge, even though they are still behind the frontier closed-source models (e.g. GPT-4o). We will include this comparison in Figure 5 for clarity.
>
>
> **More experiments on open-source models for question type analysis:** Models show distinct gaps between answering single-mode and multi-mode questions, unlike humans. (figure 5 left)
>
> | Model                        | Cultural Single-Mode | Cultural Multi-Mode |
> |------------------------------|----------------------|----------------------|
> | llama3-8b                    | 22.0%                | 14.0%                |
> | llama3-70b                   | 31.0%                | 23.3%                |
> | llama3-1-8b                  | 39.4%                | 19.8%                |
> | llama3-1-70b                 | 56.0%                | 30.2%                |
> | llama3-1-405b                | 53.9%                | 34.9%                |
> | Aya-expense-8b               | 29.6%                | 25.6%                |
> | Aya-expense-32b              | 36.7%                | 30.2%                |
> | 30 tested models avg (Fig 4&5)| 43.4% (SD: 11.2%)    | 25.2% (SD: 8.1%)     |
> | Best performing model (GPT-4o)| 62.8%                | 43.0%                |
> | Human                        | 92.7%                | 90.5%                |
>
>
> **More experiments on open-source models for time version:** Models in the same series improve across time versions. (figure 5 right)
>
> | Model Size        | Earlier             | Later             |
> |-------------------|---------------------|-------------------|
> | Llama: 8b         | 21.4% (Llama-3)     | 38.0% (Llama-3.1) |
> | Llama: 70b        | 30.4% (Llama-3)     | 54.2% (Llama-3.1) |
> | GPT-4o            | 61.5% (20240513)    | 62.0% (20240806)  |
> | GPT-4             | 59.8% (1106-preview)| 61.6% (0125-preview) |
> | GPT-3.5-turbo     | 27.0% (1106)        | 32.8% (0125)      |

---

> ### Author Response · Authors · 2024-11-23
> **Response to Reviewer NUgQ – (3/3)**
>
> > Lack of discussion on how to grow and develop the benchmark. For culture knowledge benchmarks like this one, it might strengthen the paper if the authors provide sufficient and convincing discussion on how the community or any plans to grow and develop the benchmark, given its small number of test instances. This can be in the form of improving the framework in Figure 1. Without this, the benchmark will only cover a snapshot of cultures of different geographical areas which, again, makes its real-life usage questionable.
>
> We understand that developing a representative cultural benchmark needs continued efforts on collecting the dataset through public engagement. We have future plans to grow and develop CulturalBench with the community. The following is our future plan:
>
> 1. **Public gamified platform development.** While the current platform is ReactJS-based (i.e. a responsive web-app), we have developed a newer version that has new gamification elements based on Unity Engine, typically used for creating games. We hope our improved gamified platform can provide intrinsic incentives to encourage users to co-create questions with AI, similar to how they interact with FoldIt [8], a game for crowdsourcing solutions for protein folding problems, associated with the Nobel Prize for Chemistry 2024 contribution.
>
> 2. **Attract users** to the platform above, such as through social media, organizing a global competition where users try to generate as many cultural knowledge questions (similar to Hack a Prompt contest [9]), or collaborating with organizations working in less representative regions (e.g. Oceania and Africa). The goal is to improve representation of cultures that are otherwise hard to capture.
>
> 3. **Develop and release the future versions of CulturalBench.** Some versions may be hosted privately to avoid model contamination by accidentally training on it, similar to SEAL Bench [10].
>
> Currently, we are in the process of developing and testing our platform with initial test users. We went into public testing of our platform a few months ago to gather our first group of initial users’ feedback. Unfortunately, we cannot share its link due to ICLR anonymity principle. We believe that building a cultural knowledge benchmark is a collective effort for researchers and people around the world – and CulturalBench (v1) provides a good starting point.
>
> > Question: How is a binary question format classified as hard when there are more chances of a model picking either one correct from the two choices? I believe I’m missing something because this is not clearly explained in the paper. If a benchmark is hard/challenging, the first thing that comes to me is something similar to MMLU-Pro (https://arxiv.org/abs/2406.01574) where there are greater than 5 (max 10) choices.
>
> Thank you for asking about the CulturalBench-Hard binary setting. We evaluated the model at question level. The model has to answer all four binary statements with the right label (True/False) to count as “Correct”.
>
> For instance, a question in CulturalBench-Hard:
>
> 1. Question: What do Singaporeans usually use to reserve seats?
>
> Answer: Tissue
>
> Is this answer true or false for this question?
>
> 2. Question: What do Singaporeans usually use to reserve seats?
>
> Answer: Wallet
>
> Is this answer true or false for this question?
>
> 3. Question: What do Singaporeans usually use to reserve seats?
>
> Answer: Sunglasses
>
> Is this answer true or false for this question?
>
> 4. Question: What do Singaporeans usually use to reserve seats?
>
> Answer: Book
>
> Is this answer true or false for this question?
>
> The probability of getting this binary-setting question correctly by randomly guessing is (0.5)**(4) = 6.25% (= the random baseline in Figure 4). On the other hand, a random guess in CulturalBench-Easy has a probability of 1/4 = 25% as the model only has to choose an answer out of the four options, following the typical multiple-choice-question format. We understand why the reviewer might have been confused. We can change the naming of CulturalBench-Hard into CulturalBench-Binary if it is more clear. Feel free to share any other naming ideas as well.
>
> Reference:
>
> [1] https://arxiv.org/abs/2109.07958 TruthfulQA: Measuring How Models Mimic Human Falsehoods
>
> [2] https://huggingface.co/spaces/open-llm-leaderboard-old/open_llm_leaderboard
>
> [3] https://arxiv.org/abs/2311.12022 GPQA: A Graduate-Level Google-Proof Q&A Benchmark
>
> [4] https://openai.com/index/learning-to-reason-with-llms/
>
> [5] https://www.anthropic.com/claude/sonnet
>
> [6] https://arxiv.org/abs/2009.03300 MMLU
>
> [7] https://arxiv.org/abs/2407.21783 The Llama 3 Herd of Models
>
> [8] https://pmc.ncbi.nlm.nih.gov/articles/PMC2956414/ Predicting protein structures with a multiplayer online game
>
> [9] https://arxiv.org/abs/2311.16119 Ignore This Title and HackAPrompt: Exposing Systemic Vulnerabilities of LLMs through a Global Scale Prompt Hacking Competition
>
> [10] https://scale.com/blog/leaderboard

---

> > ### Comment · Reviewer_NUgQ · 2024-11-24
> > **Acknowledgment of response**
> >
> > This is to acknowledge that I have read the author/s response to my reviews and other reviewers' feedback.
> >
> > I appreciate the additional experiments done by the author to shed light on the issues I have raised. While the author/s has clarified that the benchmark prioritizes quality (breadth) over quantity (depth), I still feel it has to strike a balance between the two for the benchmark to be usable. If you look at the statistics in Appendix B, test instances are mostly between 7-25, which I would not be confident to use to leverage cultural understanding of LLMs. This concern has also been raised by other reviewers.  Increasing this quantity of test instances would definitely change the LLMs' performances, thus, there is expansion needed for the benchmark.
> >
> > I will be keeping my score this time.

---

> > > ### Author Response · Authors · 2024-11-29
> > > **Response to reviewer NUgQ**
> > >
> > > We thank the reviewer for their appreciation and raising their concerns.
> > >
> > > We agree that the best case would be to provide a benchmark that is both large and has high quality. However, given limitations on the budget we have for annotations, we had to make trade-offs between quantity and quality. One such tradeoff that we had to make was to allocate a part of our budget to human verifications (i.e. having five annotators answer each question to see if there is a consensus answer). Another tradeoff we had to decide was to remove close to half of the initial samples which did not have a consensus answer. If we had to optimize for quantity instead of quality, we might get around 5000 samples using the same budget.
> > >
> > > As we state in Table 1, there are many other cultural benchmarks that prioritizes quantity, but they have yet to be substantially used by LLM providers (such as Anthropic, OpenAI and Meta) . We saw that LLM-providers typically value datasets that are of high quality even if they are relatively small (TruthfulQA with 1.2k samples and GPQA Diamond with less than 200 samples) and hence we saw a research gap for a high-quality cultural knowledge benchmark, which are valuable for LLM-providers even if the benchmark only has 1.2k samples.
> > >
> > > Another consideration is that we only measure and report LLM performance by sub-continental regions (e.g. East Asia, South Europe) as in Figure 6 and hence the limited number of samples per country in Appendix B is less relevant toward the reliability of the benchmark in capturing the cultural knowledge per sub-continental region (which contains several countries). We will add a further disclaimer in the paper warning users that they should not use the accuracy on a per-country basis in light of your comment.

---

### Official Review · Reviewer_Cyqx · 2024-11-01

**Soundness:** 2
**Presentation:** 3
**Contribution:** 2
**Rating:** 5
**Confidence:** 4

**Summary:**

The authors have created a cultural knowledge benchmark, CulturalBench, through AI assisted interactive red-teaming.
The major contribution I would consider to be relevant to HCI aspects, of how they setup and conducted the benchmark curation using human annotators and with AI assistance.

I believe such work fits better in HCI focused conferences.

Some other recent work (such as CultureBank) leveraged social networks (reddit, twitter etc.) to curate similar datasets and benchmarks. One can argue that collecting such data indirectly provides wider representation, especially in multicultural regions, which the authors of this paper briefly mention but dont go into much details. I.e., when thinking about culture knowledge of a country/region that is largely multi-cultural, which aspects play a large role or how those multiple identities (county of origin, ethnicity, religion, current place of residence) interplay to define what is the 'culture' of this region?

**Strengths:**

Devised an effective AI assisted framework for engaging human annotators in curation of cultural knowledge benchmarks.
The paper is well written and the AI assisted framework holds promise to increase annotators involvement/engagement.

**Weaknesses:**

The main focus of this paper is in describing the human annotators setup with AI assisted implementation.

It lacks some of the comparison with other cultural knowledge benchmarks (such as CultureBank) that were generated using other approaches and as such, lacks the discussion of how this approach is better and what are the limitations comparing to others.

As alluded by the authors, no fine-grained (and I would add, no consideration for multiplicity of identities) cultural classification - which is mentioned to be a limitation of the annotation platform used -- that may have non-neglected impact on the resulted benchmark.

**Questions:**

What was the approach taken to represent accurately multi-cultural countries? If the annotators were from specific country but not part of a specific culture (being it ethnical, religious or other), was that taken into consideration?

How about other multiplicity of identities which may have an impact on how one perceives ones culture?

Would be good to include detailed description of annotator selection criteria, particularly for multicultural countries.
I would suggest to include discussion on how the authors accounted for within-country cultural diversity in their data collection and validation process, including how they addressed the interplay of multiple identities (e.g., country of origin, ethnicity, religion, current place of residence) in defining the 'culture' of a region.

Can you include a detailed comparison between your approach and methods like CultureBank that use social media data? Discussing potential advantages and limitations of each approach for capturing cultural diversity, especially in multicultural regions

---

> ### Author Response · Authors · 2024-11-23
> **Response to Reviewer Cyqx – (1/3)**
>
> > I believe such work fits better in HCI focused conferences.
>
> > The major contribution I would consider to be relevant to HCI aspects, of how they setup and conducted the benchmark curation using human annotators and with AI assistance.
>
> > The main focus of this paper is in describing the human annotators setup with AI assisted implementation.
>
> We thank the reviewer for appreciating our robust and effective data curation pipeline. We agree that the AI-assisted data generation pipeline we designed is one novel contribution that could bring inspiration to the community. However, that is a secondary contribution of the paper, which we only devoted one section to (Section 3.1 at Lines 216 - 230).  The **main focus** of the paper is to introduce CulturalBench as a robust, diverse and challenging dataset can effectively measure the cultural knowledge of models.
>
> As such, we believe that our contribution aligns well with the goals of the ICLR conference towards "**publishing cutting-edge research on all aspects of deep learning used in the fields of artificial intelligence, statistics and data science, as well as important application**", especially under the Primary Area of Datasets and Benchmarks.
>
> To introduce CulturalBench, we
> 1. Demonstrated how CulturalBench mitigates limitations of existing benchmarks on culture knowledge (Section 2)
> 2. Collected the dataset using the AI-assisted data generation pipeline as well as other methods to improve data quality (Section 3)
> 3. Analyzed the distribution of topics within the dataset (Section 4)
> 4. Measured the performance of 30 different models with analysis across multiple model axes (Section 5)
>
> > It lacks some of the comparison with other cultural knowledge benchmarks (such as CultureBank) that were generated using other approaches and as such, lacks the discussion of how this approach is better and what are the limitations comparing to others.
>
> We appreciate all work done on improving the cultural understanding of models including CultureBank [1]. However, due to space constraints, we were only able to compare the strength and limitations of our approach relative to four most related cultural benchmarks in Table 1. CultureBank is not easily comparable with CulturalBench or the other four recent cultural benchmarks (Candle [2], CulturalAtlas [3],  Nomad [4] and Blend [5]) for the following reasons:
>
> 1. **Evaluation format:** While CulturalBench and the four compared benchmarks above are in the format in which the questions contain ground truth answers that can be easily verified. These formats include both Multiple Choice Questions as well as True/False statements. On the other hand, CultureBank evaluation is not easily verifiable as the evaluated LLM is expected to generate an open-ended answer. In order to judge whether the generated answer is suitable, it involves an LLM-as-a-Judge (GPT-4) in the loop to evaluate whether the answer entails a relevant cultural descriptor. As a result, the resulting entailment score in CultureBank is not directly comparable with the “accuracy-like” metrics reported in CulturalBench and the other four benchmarks.
>
> 2. **Granularity of Culture:** While CulturalBench and the four compared benchmarks above generally contain granularity at the level of countries, CultureBank focuses more on cultural groups (e.g. American, Californian, Asian American, international student, as shown in Table 1 of CultureBank paper [1]). While some of these cultural groups are countries (e.g. American), other cultural groups are relating to other aspects of social identity, which while valuable to study, can be difficult to directly compare to the focus on countries reported in CulturalBench and the other four benchmarks.
>
> We will clarify the above discussion in our paper. We have discussed benefits and limitations of different approaches in Section 2 and will extend this discussion (to CultureBank) in our response to the next question.
>
> Reference:
>
> [1] https://arxiv.org/abs/2404.15238 CultureBank: An Online Community-Driven Knowledge Base Towards Culturally Aware Language Technologies
>
> [2] https://arxiv.org/abs/2210.07763 Extracting Cultural Commonsense Knowledge at Scale
>
> [3] https://arxiv.org/abs/2402.09369 Massively Multi-Cultural Knowledge Acquisition & LM Benchmarking
>
> [4] https://arxiv.org/abs/2404.12464 NormAd: A Framework for Measuring the Cultural Adaptability of Large Language Models
>
> [5] https://arxiv.org/abs/2406.09948 BLEnD: A Benchmark for LLMs on Everyday Knowledge in Diverse Cultures and Languages

---

> ### Author Response · Authors · 2024-11-23
> **Response to Reviewer Cyqx – (2/3)**
>
> > Can you include a detailed comparison between your approach and methods like CultureBank that use social media data? Discussing potential advantages and limitations of each approach for capturing cultural diversity, especially in multicultural regions
>
> We agree that it could be beneficial to discuss the benefits and limitations for using different sources of data for constructing cultural knowledge benchmarks:
>
> **Social Media Data (e.g. TikTok and Reddit used in CultureBank [1]):**
>
> Benefits:
> 1. Large sources of data allow for diverse and real-life topics and discussions from users.
> 2. Low financial cost as social media APIs can be assessed freely using research program (e.g. Reddit / TikTok)
>
> Limitations:
> 1. Contains representational biases. For instance, certain demographics such as males aged 18-29 from English-speaking countries (United States, United Kingdom, Canada, Australia) are substantially over-represented on Reddit [2]. Cultural knowledge mined from participants from such demographics might not represent those from these cultures in general.
>
> **Web resources (e.g. Wikipedia used in CultureAtlas [3]):**
>
> Benefits:
> 1. Large sources of documents are organized in a hierarchical structure, allowing for easy and systematic extraction of relevant cultural information.
> 2. Low financial cost as these web resources are typically openly accessible.
>
> Limitations:
> 1. Large Language Models are likely to have been trained on such web-resources as they are commonly used as LLM pre-training data (e.g. CommonCrawl and Wikipedia) [4]. Because LLMs were already exposed to such data, benchmarks constructed from web-sources tend not to pose sufficient challenge for models. For instance, LLMs (specifically ChatGPT-3.5) can achieve 93.1% on CultureAtlas [3].
>
>
> **Human-AI Collaborative Annotation (e.g. our CulturalBench):**
>
> Benefits:
> 1. Enables the collection of challenging and high-quality benchmark questions. Challenging questions can be collected because humans have real-time access to an AI Verifier to “test” questions they create, which can discourage easy questions. High-quality questions are collected by having multiple humans (n=5) check each question after it was first created, in order to filter out unsatisfactory or ambiguous questions.
> 2. Even representation of demographics (including age, gender and region) in collected benchmarks because the dataset collection platform allows the setting of recruitment criteria on each of these demographic aspects.
>
> Limitations:
> 1. High financial cost that incurred in paying human annotators. This is a one-time cost that will not have to be repeated and hence the “amortized” cost across all users of the collected dataset is relatively low.
>
> **Overall**
>
> Each source of data has its own set of benefits and limitations and therefore, we believe that the community needs to utilize a diversity of data sources in order to pursue different research directions. In this work, our primary lies in curating a robust, diverse and challenging benchmark for measuring cultural knowledge of LLMs, inspired by the creation of similar knowledge-intensive benchmarks in other areas such as TruthfulQA [5] and GPQA [6]. Hence, we found Human-AI Collaborative Annotation to be most suitable.
>
> Reference:
>
> [1] https://arxiv.org/abs/2404.15238 CultureBank: An Online Community-Driven Knowledge Base Towards Culturally Aware Language Technologies
>
> [2] https://www.nature.com/articles/s41586-021-04167-x Quantifying social organization and political polarization in online platforms
>
> [3] https://arxiv.org/abs/2402.09369 Massively Multi-Cultural Knowledge Acquisition & LM Benchmarking
>
> [4] https://arxiv.org/abs/2305.13169 A Pretrainer's Guide to Training Data: Measuring the Effects of Data Age, Domain Coverage, Quality, & Toxicity
>
> [5] https://arxiv.org/abs/2109.07958 TruthfulQA: Measuring How Models Mimic Human Falsehoods
>
> [6] https://arxiv.org/abs/2311.12022 GPQA: A Graduate-Level Google-Proof Q&A Benchmark

---

> ### Author Response · Authors · 2024-11-23
> **Response to Reviewer Cyqx – (3/3)**
>
> > As alluded by the authors, no fine-grained (and I would add, no consideration for multiplicity of identities) cultural classification - which is mentioned to be a limitation of the annotation platform used -- that may have non-neglected impact on the resulted benchmark.
>
> We agree with the reviewer that more nuanced benchmarks with various concepts of cultural identity is important. We tried to ensure a good representation of annotators from each culture by selecting annotators to ensure even representation across age, gender and region. However, at the current stage of research into cultural understanding of LLMs., collecting/requesting for further detailed PII of participants (ethnicity, religion, current place of residence) might not be easily justifiable on an IRB / ethics review board.
>
> As it stands now, there is no domain-general culture evaluation benchmark covering a diversity of countries and domains that can sufficiently challenge current LLMs. Therefore, having this dataset released can support LLM providers to start thinking about measuring cultural knowledge in addition to math, code, instruction following.  This will make a strong foundation to justify the important follow up work up for nuanced aspects of culture that the reviewer suggests, especially in terms of collecting more sensitive personal data.
>
> > What was the approach taken to represent accurately multi-cultural countries? If the annotators were from specific country but not part of a specific culture (being it ethnical, religious or other), was that taken into consideration?
> > How about other multiplicity of identities which may have an impact on how one perceives ones culture?
> > Would be good to include detailed description of annotator selection criteria, particularly for multicultural countries. I would suggest to include discussion on how the authors accounted for within-country cultural diversity in their data collection and validation process, including how they addressed the interplay of multiple identities (e.g., country of origin, ethnicity, religion, current place of residence) in defining the 'culture' of a region.
>
> We thank the reviewer’s suggestion on describing the annotator selection criteria and their concern on accurate representation of multicultural countries.
>
> In our paper, we have described our recruitment criteria in Section 3.2 (Step 2: Human Quality Check) and the process of handling more fine-grained. We can share more details and our justification:
>
> **Recruitment.** To cover a diverse range of cultures during recruitment, we used geographical locations (i.e. “region/country”) as the boundary of each culture.
>
> We set two criteria:
> 1. Nationality (the annotators have the nationality of the target region/country)
> 2. Primary residence before age 18. (the annotators spent most of their time in the target region/country).
>
> We believe these two criteria are sufficient to ensure our recruited annotators having rich cultural knowledge per each targeted region/country. Therefore, our CulturalBench can have accurate data for each region/country. Another consideration is that we limited annotators to providing 5-7 questions each, which ensured a large number (> 400) of annotators across the entire collection. This large number of annotators can increase the chance of having a diverse representation across the aspects mentioned by the reviewer (ethnicity, religion, current place of residence), given that it might otherwise be ethically questionable to use these aspects for annotator selection, as we discussed in our response above.
>
> **Questions that are specific to certain subcultures (e.g. smaller region).** After collecting questions, we sent these questions for further human verification. Each question was verified by five independent human annotators per region/country (using the same recruitment criteria). Annotators were allowed to flag the question with “I don’t have knowledge” or “This question is unanswerable” during verification. For instance, when a question of “What do visitors in Scotland often think they can hunt and eat?” was verified by five people from the United Kingdom, it received four votes on “I do not have knowledge”.  When questions were flagged by more than two times, we will review the questions to see if we can further distribute this question to specific sub-regions in the United States and United Kingdom (e.g. Texas in US and Scotland in UK). Sub-regions in other countries were not available on the annotation platform we used.

---

### Official Review · Reviewer_cBbp · 2024-11-05

**Soundness:** 2
**Presentation:** 3
**Contribution:** 2
**Rating:** 5
**Confidence:** 3

**Summary:**

This paper introduces CULTURALBENCH, a new benchmark for evaluating large language models' (LLMs) cultural knowledge across 45 global regions, including underrepresented areas like Bangladesh, Zimbabwe, and Peru. The benchmark consists of 1,227 human-written, human-verified questions spanning 17 cultural topics, such as food preferences and greeting etiquette. The authors propose two evaluation setups: CULTURALBENCH-Easy and CULTURALBENCH-Hard, which feature the same questions phrased differently to assess the models' robustness to variations in question formulation.

The paper introduces a comprehensive, diverse benchmark to assess LLMs’ cultural understanding across regions that are often neglected in existing datasets. By using two different question setups, the authors highlight that LLMs' performance is sensitive to how questions are framed, with differences as high as 27.3% observed in GPT-4o. The authors compare LLMs, including GPT-4o and Llama3-8b, against human performance. While human annotators achieve 92.6% accuracy, the best LLM (GPT-4o) reaches only 61.5% accuracy on the more challenging setup, revealing significant gaps in cultural knowledge. The paper identifies specific shortcomings in LLMs, particularly in regions like South America and the Middle East, as well as difficulties in handling questions with multiple valid answers.

**Strengths:**

The authors collect a culture-related question-answering dataset by using a pipeline consisting of three steps: 1) Red-teaming data collection; 2) Human Quality Chek 3) Filtering. The proposed datset is verified by five annotators. There are several merits of this dataset:

1. The dataset is divided into CulturalBench-easy and CulturalBench-hard, which are more differentiable than current culture-related datasets, covering more diverse topics and more challenging questions.

2. The model's performance on CULTURALBENCH-Hard is far behind human performance, where GPT-4o achieves the best performance.

3. The authors propose some interesting analysis on different providers' LLMs on questions from different regions. For example, they find that models perform better in questions relating to North America, South Asia, and West/South Europe than South America, East Europe, and the Middle East. Meanwhile, Qwen-2-72B and Mistral Large do not perform well in answering their own region's cultural knowledge.

**Weaknesses:**

My major concern with this work is the form of the multiple choice question. As many researchers agree and argue before, LLMs usually show unstable and unpredictable performance when are required to answer multiple choice questions. Why not design some generation tasks to evaluate LLM's capability towards culture-related questions?

While culture-related question-answering problem is not novel at the moment given the previous similar work, including CultureLLM, CultureBank, and SeaLLM. I am wondering whether the authors could propose more diverse data types, not limited to the multiple-choice format. How do your multiple-choice formats compare to or improve upon the formats used in these works?

In addition, the experimental results are not counterintuitive enough. For instance, GPT-4o leads in most regions, Models score lower in questions relating to South America, East Europe, and the Middle East, etc. More in-depth analyses are expected to be done.

**Questions:**

1. Have you considered other question formats beyond the multiple choice question answering?

2. Have you considered evaluating the self-knowledge of LLMs when answering culture-related questions. As shown in Line 511 to Line 513, the authors provide two extra options: “I don’t have knowledge” and “This question is unanswerable” – to enable annotators to indicate when they cannot provide a response. However, we have not observed the comparison between LLMs and humans in terms of the self-knowledge.

---

> ### Author Response · Authors · 2024-11-23
> **Response to Reviewer cBbp – (1/4)**
>
> > My major concern with this work is the form of the multiple choice question. As many researchers agree and argue before, LLMs usually show unstable and unpredictable performance when are required to answer multiple choice questions. Why not design some generation tasks to evaluate LLM's capability towards culture-related questions?
>
> > Have you considered other question formats beyond the multiple choice question answering?
>
> We recognize the limitation of Multiple Choice Question format in that it simplifies the task to have the correct answer to be among (typically) four options, making it somewhat easy to guess the correct answer (~25% chance) even if they cannot otherwise provide the correct answer. Based on this observation, we propose a new **binarized** multiple choice question setup (i.e. CulturalBench-hard) where the model needs to consider whether each option is individually True or False. We only consider the model as able to get the question correct if it identifies three options as False and one option (the annotated answer) as True.
>
> Early on in our research design, we did consider various question formats such as free-text generation. One particular work that inspired our eventual choice was TruthfulQA [1], which compared different formatting of posing questions relating to factual knowledge understanding (to LLMs). Specifically, they compared the multiple choice format with a free-text generation format. While scoring for multiple choice questions is relatively straightforward, the researchers found that the correctness of the free-text generation answers cannot be effectively measured using overlap based metrics (e.g. ROUGE, BLEU with respect to Reference answers). This is because answers can be correct without mentioning keywords by using synonyms or paraphrases. To solve this issue, the researchers used a separate LLM (GPT-3) trained to judge if a generated answer is similar to the reference answer. Fast-forward 3 years, TruthfulQA is now widely known (with over 1,200 citations) but most researchers only use the multiple choice format, for instance the HF Open LLM v1 Leaderboard [2] which contains hundreds of models. We believe this is because having a separate LLM compare if the generated-answer is similar to the reference answer is an error-prone process due to inherent biases of the judge-LLM [3].
>
> Because our goal is to encourage LLM providers to start benchmarking a model’s cultural knowledge, we want to collect the dataset in a format that will encourage and facilitate their use - in this case the Multiple Choice Question (MCQ) format seemed most suitable. The popular adoption of other knowledge-intensive benchmarks in MCQ format such as MMLU [4] and GPQA [5] further convinced us that the MCQ format is most convenient for use by LLM providers as a start.
>
> Reference:
>
> [1] https://arxiv.org/abs/2109.07958 TruthfulQA: Measuring How Models Mimic Human Falsehoods
>
> [2] https://huggingface.co/spaces/open-llm-leaderboard-old/open_llm_leaderboard
>
> [3] https://arxiv.org/abs/2306.05685 Judging LLM-as-a-Judge with MT-Bench and Chatbot Arena
>
> [4] https://arxiv.org/abs/2009.03300 Measuring Massive Multitask Language Understanding
>
> [5] https://arxiv.org/abs/2311.12022 GPQA: A Graduate-Level Google-Proof Q&A Benchmark

---

> ### Author Response · Authors · 2024-11-23
> **Response to Reviewer cBbp – (2/4)**
>
> > While culture-related question-answering problem is not novel at the moment given the previous similar work, including CultureLLM, CultureBank, and SeaLLM. I am wondering whether the authors could propose more diverse data types, not limited to the multiple-choice format. How do your multiple-choice formats compare to or improve upon the formats used in these works?
>
> While many researchers explored cultural question answering with LLMs (e.g. CultureLLM designed a pipeline of collecting data to finetune the model), we found that the current cultural benchmarks cannot effectively test the models’ cultural understanding. In Table 1, we found that existing LLMs perform at more than 80% accuracy for each of these benchmarks, meaning that it’s no longer a useful target for LLM builders to use for assessing cultural knowledge. Therefore, there is an urgent need to create a more challenging dataset. We find that most of this challenge comes from the content of the questions rather than the format, as discussed in the response to the previous question.
>
> Perhaps an analogy would be useful here.
>
> When we evaluate the math capabilities of a student, we ask them simple arithmetic questions early on (e.g. 2+5=? Or 3 x 4 =?). However, as they master these simple topics, we need harder questions such as algebra (2a + 1 = 5, a= ?) and even calculus (d/dx (5x +1) = ? ) to evaluate them better rather than to use the same arithmetic questions since all students can answer arithmetic easily. Even for very advanced math evaluation (e.g.  SAT or GRE), we can use multiple choice question format because it is much easier to automatically grade.
>
> Similarly, the analogy can be applied to evaluating the world knowledge of LLMs. We started from MMLU [1] (high school / college level questions)  but as models reach more than 80% performance (see [2]), we need more challenging benchmarks such as GPQA [3] (graduate-level questions), despite keeping the same MCQ format.
>
> As current cultural knowledge evaluation works (such as CultureLLM) are starting to be saturated by current LLMs, we want to support the community by building a more challenging dataset, which can more effectively assess cultural knowledge among LLMs.
>
> Reference:
>
> [1] https://arxiv.org/abs/2109.07958 TruthfulQA: Measuring How Models Mimic Human Falsehoods
>
> [2] https://paperswithcode.com/sota/multi-task-language-understanding-on-mmlu
>
> [3] https://arxiv.org/abs/2311.12022 GPQA: A Graduate-Level Google-Proof Q&A Benchmark

---

> ### Author Response · Authors · 2024-11-23
> **Response to Reviewer cBbp – (3/4)**
>
> > In addition, the experimental results are not counterintuitive enough. For instance, GPT-4o leads in most regions, Models score lower in questions relating to South America, East Europe, and the Middle East, etc. More in-depth analyses are expected to be done.
>
> We find the claim that the “experimental results not being counterintuitive enough” by the reviewer to raise interesting questions. Whether a result is not counterintuitive enough, depends heavily on what intuitions a reader has. In this particular case, the reviewer seems to suggest that they expect GPT-4o to lead in most regions. It is not clear why this should be the case, especially since OpenAI GPT-4o does not serve certain markets (e.g. China in East Asia) and hence might not be expected to do well compared to LLMs developed by Chinese enterprises and trained on data more related to local culture.
>
> It’s possible that the reviewer developed this intuition based on the assumption that GPT-4o is generally known to be a strong LLM and hence found it not surprising that it does well in this benchmark too. However, GPT-4o does not always exceed performance of other LLMs (like Llama 3.1 405B and Claude 3.5 Sonnet) on benchmarks such as Chatbot Arena [2] and Aider Bench [3]. This very observation that different people have different intuitions about the cultural knowledge of various LLMs precisely demonstrates why we need a benchmark like CulturalBench to guide our intuitions with empirical results.
>
> The same observation by the reviewer that “GPT-4o still leads in most regions” given that other model providers have specialized in some cultures was actually **counterintuitive (and alarming)** to our research team. To better understand this, we conducted comparative analysis on different models for their related regions (see Section 5.3).
>
> In our CulturalBench-Hard, Qwen-2-72b scores 50.7% on East Asia, while GPT-4o achieves 61.4%. Mistral Large underperforms in West Europe (48.9%) compared to GPT-4o (54.3%). We also newly tested the Aya models (suggested by reviewer NUgQ) since they focused on training multilingual capabilities. We found that Aya-32b [1] scores 36.3% and cannot beat other open-source models (e.g. Llama-3.1-70b with 54.2%).
>
> These results (including the fact that GPT-4o scores the highest for most of the regions “unsurprisingly”) show a **pressing need** for the community to rethink how we can best train models to acquire cultural knowledge.
>
> Reference:
>
> [1] https://huggingface.co/CohereForAI/aya-expanse-32b
>
> [2] https://lmarena.ai/?leaderboard
>
> [3] https://aider.chat/docs/leaderboards/

---

> ### Author Response · Authors · 2024-11-23
> **Response to Reviewer cBbp – (4/4)**
>
> > Have you considered evaluating the self-knowledge of LLMs when answering culture-related questions. As shown in Line 511 to Line 513, the authors provide two extra options: “I don’t have knowledge” and “This question is unanswerable” – to enable annotators to indicate when they cannot provide a response. However, we have not observed the comparison between LLMs and humans in terms of the self-knowledge.
>
> The main reason why we provide these two extra options is to flag questions unsuitable to be included in to the benchmark dataset instead of evaluating the self-knowledge of humans/llms. Specifically, unsuitable questions refer to questions that they cannot answer (I don’t have knowledge) or constructed in a problematic way (This question is unanswerable). For the first type, it could be questions that are in other subcultures (e.g. Scottish culture in the United Kingdom -- “What do visitors in Scotland often think they can hunt and eat?”). We will send this question to a new human annotator and set a more specific criteria when applicable (e.g. recruiting individuals living in Scotland). For the second type, it could be questions that are too generic or have no right answer (e.g. Which food do United States people like the most). We will also send this question to a new human annotator. If this question has been flagged more than 2 times, we will withdraw this question from our question pool. **With such a human verification pipeline by people who have the specific cultural knowledge, we hope to build a robust benchmark that contains suitable questions for each culture.**
>
> Nonetheless, this is an interesting question around evaluating the self-knowledge of LLMs when answering culture-related questions. Our understanding of self-knowledge evaluation is primarily based on the definition in [1]. Relating this back to culture knowledge, we are not sure how modeling self-knowledge in this context might be useful in a benchmark (our main objective). For instance, if a model gets 60% of questions correct, 20% wrong and 20% “I don’t have knowledge” - it’s not clear how it should be compared with another model that gets 60% correct and 40% wrong or even a third model with 75% correct and 25% wrong. In the extreme case, a model can even choose “I don’t have knowledge” for all questions.
>
> Moreover, we are not sure why it might be useful to compare LLMs and humans in this context. Using the question - “What do visitors in Scotland often think they can hunt and eat?”, maybe 3 out of 10 humans living in the United Kingdom do not know the answer to this (while 7 humans with knowledge of Scottish practice do). Assuming this proportion is representative of all CulturalBench questions, “I don’t have knowledge” will be 0.3 for humans. However, in the current way that many people use LLMs (as general helpful assistants), it is not that useful for them to answer ‘I don’t have knowledge’ 30% of the time. In this sense, we expect LLMs to have cultural knowledge beyond what an individual human has and therefore comparing humans and LLMs in this aspect is not informative.
>
> Reference:
>
> [1] https://arxiv.org/abs/2207.05221 Language Models (Mostly) Know What They Know

---

> > ### Comment · Reviewer_cBbp · 2024-11-24
> > **Thanks for your response.**
> >
> > Thanks for such a careful response and for proposing many open questions around the topic of cultural knowledge. I do agree with some updated opinions, such as the MCQ format following the TruthfulQA. However, the key to a research paper could be its novelty and unique contribution based on previous attempts. Given the existing findings from CultureLLM, CulturePark and CultureBank, I do not see a significant difference between CultureBench with these works, except for the incremental contributions. Hence, I would like to maintain my score.

---

> > > ### Author Response · Authors · 2024-11-29
> > > **Response to reviewer cBbp**
> > >
> > > We thank the reviewer for their feedback! We are unclear about why the reviewer does not see significant differences between this work and CultureLLM, CulturePark and CultureBank and we would love for the reviewer to elaborate.
> > >
> > > The contributions of this work (CulturalBench) are:
> > >
> > > 1. We propose a Human-AI collaboration platform for collecting diverse, robust and challenging cultural knowledge datasets.
> > > 2. We use the platform to curate a dataset with 1,227 samples across 45 cultures and 17 topics.
> > > 3. We benchmark 30 different models using the dataset, analyzing their performance across the axes of LLM-provider, time of release and sub-continental regions (e.g. South Asia).
> > >
> > > On the other hand,
> > >
> > > 1. **CultureLLM** trains a LLM on synthetically-generated data and evaluates it in safety-related tasks relating to “offensive language, hate speech, stance, toxicity, threat, bias, abusive”
> > >
> > > 2. **CulturePark** is an LLM-powered multi-agent communication framework for synthetically generating training data to fine tune LLMs which are then evaluated on cultural moderation, cultural alignment and cultural education.
> > >
> > > 3. **CultureBank** is a knowledge base built upon users’ self-narratives with cultural descriptors sourced from TikTok and Reddit, focussing more on social identity groups (e.g. American, Californian, Asian American, international student), as shown in Table 1 of CultureBank paper.
> > >
> > >
> > > Specific areas of difference:
> > >
> > > 1. Our work focuses on cultural knowledge but CultureLLM and CulturePark do not. CultureBank focuses on social identity groups (Asian Americans, international students, Californians) while our work focuses on much bigger regions (e.g. North America) - please see more about comparison of our work with CultureBank in our response to reviewer Cyqx. We found four other works that are most related to our work in Table 1 and carefully discussed how our work is differentiated from them.
> > > 2. While CultureLLM, CulturePark and CultureBank all only depend on either LLM synthetic data or internet resources, our work introduces a Human-AI collaboration platform and uses that platform to curate a dataset. The involvement of humans is critical as humans can provide ideas for generating questions and verify if the answers are correct (by multiple annotators), in ways that LLMs alone are able to do well in. This approach means that the constructed dataset can be more diverse, robust and challenging.
> > > 3. Our work is focused on building an evaluation benchmark while the CultureLLM, CulturePark and CultureBank are primarily about training LLMs. The consequence is that our evaluations included a much wider variety of models (30 different models as compared to <10 in each of CultureLLM, CulturePark and CultureBank), providing insight into many different models across different axes (LLM provider, time of release and sub-continental regions).

---

> > > > ### Public Comment · ~Cheng_Li33 · 2024-11-29
> > > >
> > > > As the author of CultureLLM and CulturePark, I've followed this paper since its ICLR submission. While this work addresses culture and LLMs broadly, it differs from our work in CultureLLM and CulturePark. Their focus is on developing an evaluation benchmark, whereas we concentrate on fine-tuning culture-specific models. Though several reviewers referenced our work (which we appreciate!), I want to clarify that this paper shouldn't be challenged based on perceived similarities to our research.

---

### Official Review · Reviewer_M6Pe · 2024-11-06

**Soundness:** 2
**Presentation:** 3
**Contribution:** 2
**Rating:** 5
**Confidence:** 5

**Summary:**

This study proposes 1,227 human-written and human-verified questions to asses cultural knowledge of LLMs covering 45 global regions, and 17 diverse topics. Each question is verified by five independent annotators.

**Strengths:**

- Coverage of 45 regions associated with different culture
- Evaluated using 30 current LLMs.
- Details experimental results with different analysis

**Weaknesses:**

- Number of questions are relatively low compare to any other recent datasets. While it covers more countries, but number of questions per country is low (Appendix B, Table 2). For example, how do you justify the performance by just asking 7 questions (e.g., Canada)? This performance evaluation can have both effects, either it is underestimating of overestimating the performance. As a rule of thumb, a sample size of at least 30 is considered the minimum for statistical analysis due to the Central Limit Theorem.
- L161: "but did not verify the final questions by humans" --> Is this necessary when question is already written by human? Do you mean verifying by other annotators.
- L203: "These non-human written benchmarks" --> this is a contradictory statement. This study also used LLMs in the process, which makes the authors' criticism of prior work for doing the same seems contradictory. In addition, this study criticizes prior work "Over-reliance on Web Sources -- scraped web sources may be used during models pretraining". This whole question formulation process is based on LLMs knowledge, which is inherently knowledge from web sources.
- L206-208: Low performance does not necessary mean challenging.
- L329: GPT-4o-mini is used for the classification of questions into topics. It could be have done manually. This categorization is subjective in nature and requires a guideline to follow. For example, one can argue that "how to greet people" is a daily life activity. One of the major claims/contributions is the breadth of topic coverage, which has been addressed inadequately.
- I did not find any prompt in Appendix C.
- What was the reason to limit to "most 3-7 questions"? Is there any reference that culture can be defined by 5-7 questions?
- L333-336: The discussion seems arbitrary. Annotators ask question based on their knowledge. If number of annotators are less, meaning we are limiting the background knowledge, also limiting the diversity of the questions.
- This study is highlighting "discovery approach" very much. It is important to justify why it is beneficial.
- While it is an study about cultural knowledge, therefore, one would expect that dataset will be in native language, however, this study is focused on English.

- By looking at the resource, it seems that question is generic, can be applicable to different countries by replacing country. See first example here. For the second example, answer varies location to location within the culture.

23,23,"In Bangladesh, which festival celebration style is most likely appropriate for expressing the collective cultural identity?",Hosting a potluck where each individual brings a dish from their own ethnic background.,Celebrating only the most widely recognized national holiday.,Selecting one culture each year to highlight and exclusively celebrate its festivals.,Ignoring all cultural festivals to maintain neutrality.,B,Bangladesh

94,94,"In Bangladesh, what is a customary action before stepping into someone's home?",Light incense at the entrance,Remove your shoes,Offer to cook a meal,Clap your hands twice,B,Bangladesh

**Questions:**

- What does it mean to be "less represented", as per as online digital content representation languages are typically categorized as low, medium and high resource languages.
- What is the purpose of CULTURALBENCH-Hard?
- L171: How many topics initially discovered by the annotators? As it was open-ended, one can assume that number of topics could be even more than 17 topics
- Which model is used for brainstorming?
- "culturally relevant scenarios" --> Was it completely open? Were there any instructions like what it means to be culturally relevant scenarios? Example in Figure 10 is not clear. As an annotator how should one start with it?
- Answers often come in different forms such short or long for the same question. Did you observe any such? How did you manage that? It may come from multiple annotators. What values (e.g., temperature) did you choose on the LLM while generating question and answer? Was it same for all annotators.
- L348: For the evaluation, did the model always generated correct answer? It is possible that it produced extra tokens. How did you deal with that?
- L376: How did you compute human performance - 92.4%?

---

> ### Author Response · Authors · 2024-11-23
> **Response to Reviewer M6Pe – (1/5)**
>
> We appreciate the reviewer’s comments and feedback.
>
> > Weakness: Number of questions are relatively low compare to any other recent datasets. While it covers more countries, but number of questions per country is low (Appendix B, Table 2). For example, how do you justify the performance by just asking 7 questions (e.g., Canada)? This performance evaluation can have both effects, either it is underestimating of overestimating the performance. As a rule of thumb, a sample size of at least 30 is considered the minimum for statistical analysis due to the Central Limit Theorem.
>
> > Weakness: L161: "but did not verify the final questions by humans" --> Is this necessary when question is already written by human? Do you mean verifying by other annotators.
>
> CulturalBench is a small but high-quality dataset (with 1.2k samples), in which every sample has been independently validated by 5 humans. It is important to have samples to be verified by multiple human annotators (i.e. other than the question’s writer) even though our questions are written by humans for the following reasons:
>
> 1. **Some cultural questions are very subjective** and we need cultural experts (i.e. people who are from the culture) to identify what they are. For example, a rejected question - “Which food do United States people like the most? A. Burger B. Pizza C. Sushi D. Taco ” - is mostly based on personal preference and cannot be used to draw a conclusion that applies to most human annotators from the United States background. We set a stringent inclusion criteria of having at least four out of five agreeing on the correct answer to avoid including such subjective answers into the benchmark.
>
> 2. **Multiple valid answers can coexist for one cultural question.** For instance, questions like “what utensil do Chinese people usually use everyday?”, the question writer says “chopsticks” as the only correct answer. However, when other people in China (their hometown food involves porridge or soup), they may also believe answers such as “spoon” to be correct. Therefore, verification by other human annotators and allowing them to select more than one correct answer is crucial to include such diversity across different regions of a culture (See more in Line 87-95 in Introduction section).
>
> On the other hand,  slightly larger datasets (e.g. NormAd [1] with 2.6k samples) only ask 2 humans to validate up to 20% of samples while even larger datasets (e.g. CultureAtlas [2] with 10k samples) do not go through human validation. Given constraints on annotator time and budget, we chose to prioritize quality of our samples by allocating more of our resources to human validation of existing samples (rather than annotating more samples). A disclaimer is that we initially collected more than 2000 samples but chose to filter away close to half of our original collected samples post-human validation, in order to set a high bar on the quality of the benchmark dataset.
>
> For evaluation datasets, having high quality is extremely important - more so than quantity. TruthfulQA [3] is trusted by over 1, 200 research work that cites it, including HF Open LLM v1 Leaderboard [4], despite being of the same size as CulturalBench. We are also inspired by the use of rigorous human validation in data curation for GPQA Diamond [5], which results in a small dataset with fewer than 200 questions. Despite its small size, many recent works such as OpenAI o1 [6] and Anthropic Claude 3.5 Sonnet (Oct 2024 version) [7] are choosing to benchmark their models using it due to its high quality over much larger but lower quality benchmarks such as MMLU [8] (with over 15k samples).
>
> In the specific case of Canada, we did initially collect more than 30 samples per country (and planned our budget according to this), but later in the human validation process, we found some of them to be of poor quality and chose to value quality over quantity and removed more than half during our human validation process, resulting in the current situation. However, we didn't want to be blocked by this situation because countries with a low number of participants tended to be those that are well-represented in western cultures. For instance, we found that Canadian culture is very similar to US Culture due to geographical proximity of major cities (e.g. Vancouver with Seattle, Toronto with New York) and hence did not want our limited resources on cultures that are likely well represented.
>
> Reference:
>
> [1] https://arxiv.org/abs/2404.12464 NormAd
>
> [2] https://arxiv.org/abs/2402.09369 Massively Multi-Cultural Knowledge Acquisition & LM Benchmarking -- CultureAtlas
>
> [3] https://arxiv.org/abs/2109.07958 TruthfulQA
>
> [4] https://huggingface.co/spaces/open-llm-leaderboard-old/open_llm_leaderboard
>
> [5] https://arxiv.org/abs/2311.12022 GPQA
>
> [6] https://openai.com/index/learning-to-reason-with-llms/
>
> [7] https://www.anthropic.com/claude/sonnet
>
> [8] https://arxiv.org/abs/2009.03300 MMLU

---

> ### Author Response · Authors · 2024-11-23
> **Response to Reviewer M6Pe – (2/5)**
>
> > L203: "These non-human written benchmarks" --> this is a contradictory statement. This study also used LLMs in the process, which makes the authors' criticism of prior work for doing the same seems contradictory. In addition, this study criticizes prior work "Over-reliance on Web Sources -- scraped web sources may be used during models pretraining". This whole question formulation process is based on LLMs knowledge, which is inherently knowledge from web sources.
>
> > Original Line 201-203: Over-reliance on Web Sources. Existing benchmarks often rely on web sources directly such as web corpus (Nguyen et al., 2022), Wikipedia (Naous et al., 2023), and incorporated with LLMs’ generation (Rao et al., 2024; Fung et al., 2024). These non-human written benchmarks may not be challenging since …
>
> We clarify that “non-human written benchmarks” refer to the benchmarks for which no humans were involved in the question-creation or question-vetting process to ensure that the questions are of good quality.
>
> In contrast, CulturalBench questions are co-written with a Human-AI collaboration system, with all questions subsequently vetted by other humans to ensure quality. In particular, the question formulation is not based on LLMs knowledge, but instead draws from both human and LLM knowledge. To elucidate how this is the case, we would like to walk the reviewer through step 1 - Data collection on an AI-assisted red teaming platform in the data generation process (See in Figure 2 and section 3).
>
> - (i) **Initial question from human idea.** We recruit humans to use our AI-assisted platform during question creation. Human question writers first provide their idea about relevant topics related to their culture (e.g. “Singaporeans use tissues to reserve seats). Then, AI helps to elaborate upon the idea into a well-formatted multiple-choice question. (e.g. What do Singaporeans usually use to reserve seats? A. Tissue …D. book).
> - (ii) **Human editing to make questions more challenging.** Humans can edit the question and candidate options to make the question more challenging.
>
> Humans are critical in every step of the question creation, editing and verification process to ensure the question faithfully represents the cultural knowledge of the targeted region, with AI playing a “co-pilot” role to support humans.
>
> > L206-208: Low performance does not necessary mean challenging.
>
> The understanding that “low performance of LLMs on a benchmark (relative to humans)” indicates “the challenging nature of benchmarks” aligns with the interpretation by frontier LLM providers, such as OpenAI [1].
>
> For reference (see Figure 4),  Humans can do almost perfectly (i.e. 92.4%) on CulturalBench-Hard while the best performing frontier model (GPT-4o) only achieves 61.5%. If the reviewer has an alternative perspective on what a low performance might mean, please feel free to elaborate.
>
> Reference:
>
> [1] https://openai.com/index/learning-to-reason-with-llms/
>
> >L376: How did you compute human performance - 92.4%?
>
> Each question has been verified by five independent human annotators. Therefore, each question will have 5 sets of labels. After filtering to have questions with options having a majority vote (at least four out of five votes), we calculated the proportion of human votes that agree with this majority vote for all of our questions.
> For instance, question 1 with 4 out of 5 votes on option A (80%), question 2 with 5 out of 5 votes on option D (100%), question 3 with 5 out of 5 votes on option D (100%). Average among these three questions = (80% + 100% + 100%)/3 = 93.3%. For this toy instance of 3 questions, human performance will be reported as 93.3%.

---

> ### Author Response · Authors · 2024-11-23
> **Response to Reviewer M6Pe – (3/5)**
>
> > L329: GPT-4o-mini is used for the classification of questions into topics. It could be have done manually. This categorization is subjective in nature and requires a guideline to follow. For example, one can argue that "how to greet people" is a daily life activity. One of the major claims/contributions is the breadth of topic coverage, which has been addressed inadequately.
>
> > L171: How many topics initially discovered by the annotators? As it was open-ended, one can assume that number of topics could be even more than 17 topic
>
> We use a topic classification pipeline inspired by similar pipelines in Arena Hard [1] and BERTopic [2]. While we agree that classifying questions into topics can be done manually by humans, asking humans to classify topics have some disadvantages:
> This topic classification task is relatively simple and GPT-4o-mini was able to do the task well based on our spot checks. Therefore, we decided it was not worth spending annotator time on this task, and instead spent our limited annotator budget on tasks such as question generation and verification.
>
> Humans are subjective and could take substantial training before reaching high inter-annotator agreement. The example raised by the reviewer exemplifies this point because our designed schema classifies “greeting others” as under social etiquette but human annotators might disagree.
>
> Another challenge with asking humans to perform topic modeling is that many questions contain culture-specific keywords (e.g. Kutia relating to Ukraine) that annotators unfamiliar with the specific culture might not be able to effectively classify these samples.
> In addition, we believe that having LLMs do such simple tasks enables our pipeline to be more reproducible, such that researchers wishing to build similar datasets (e.g. for other cultures) can more easily build upon our approach.
>
> Reference:
>
> [1] https://arxiv.org/abs/2406.11939 Arena-Hard
>
> [2] https://maartengr.github.io/BERTopic/getting_started/representation/llm.html#selecting-documents
>
> > I did not find any prompt in Appendix C.
>
> We included the definitions of topics used in Appendix C.
> The exact prompt is:
> ```
> Based on the given question and the definitions of these topics, classify the question to the most related topic (one topic only). Definition of different topics: {definitions}.\nQuestion: {question_template}
> ```
>
> > What was the reason to limit to "most 3-7 questions"? Is there any reference that culture can be defined by 5-7 questions?
>
> The limit here refers to the number of questions that each annotator can provide, rather than the number of questions for each culture. The limit was imposed to avoid having a particular culture overly represented by a few annotators, who might inevitably bring some subjectivity in terms of their culture knowledge.
>
> > L333-336: The discussion seems arbitrary. Annotators ask question based on their knowledge. If number of annotators are less, meaning we are limiting the background knowledge, also limiting the diversity of the questions.
>
> > This study is highlighting "discovery approach" very much. It is important to justify why it is beneficial.
>
> > Original L333-336: Notably, in curating CULTURALBENCH, we observed that people from different regions focused on distinct topics. For instance, annotators from Italy and Mexico provided more questions related to Food, with 15 out of 35 questions and 13 out of 49 questions respectively. In contrast, participants from South Africa and India focused more on Religion, contributing 19 out of 58 questions and 14 out of 46 questions respectively.
>
> Our purpose of including this discussion was to show that annotators from various cultures focus on different aspects of culture, when we asked them to come up with a cultural knowledge question. This suggests that domain-specific culture datasets (such as relating to cultural cuisines) can contain substantial knowledge for cultures that emphasize on food (e.g. Italy/Mexico) while being less useful for cultures that focus less (e.g. United Kingdom). Therefore, we believe in capturing what people from each culture considers to be cultural knowledge, without pre-determining the domain they can write questions about. This discovery-based approach can empower annotators to emphasize on the aspects that they find most culturally-relevant, even if it does not fall under a common category such as food. For instance, this allowed us to include questions like “In Chinese culture, which number is likely considered unlucky?” even though it does not strictly fall into a common category.
>
> **In contrast to the reviewer’s perception, we purposefully recruited as many annotators as our budget allowed, by limiting the number of questions each annotator can create (to 5 - 7 questions/annotator),** therefore recruiting more than 400 unique annotators and improving the diversity of the questions. We will further clarify this in the paper.

---

> ### Author Response · Authors · 2024-11-23
> **Response to Reviewer M6Pe – (4/5)**
>
> > While it is an study about cultural knowledge, therefore, one would expect that dataset will be in native language, however, this study is focused on English.
>
> This was an intentional choice made by the research team for the following reasons:
>
> 1. The primary target audience of CulturalBench are researchers and developers building LLMs, who we hope can make use of CulturalBench to measure the cultural knowledge of their LLMs prior to model releases. These model developers are mostly likely to be fluent in English, and likely a few other languages but not all of the languages native to the 45 regions of interest in CulturalBench. Creating a dataset that is in English improves the interpretability and transparency, which encourages them to use it. For LLM builders serving international users, understanding how LLMs show cultural knowledge to English prompts is important (e.g. for visitors hoping to travel to a new region and wishing to learn about the culture before going).
>
> 2. Our data is collected using a Human-AI collaboration system (see Figures 1 and 2). At the start of our data collection, there was insufficient evidence to suggest that LLM used in our system (GPT-3.5-turbo, chosen due to latency considerations) demonstrates good fluency in all of the native languages among the 45 regions of interest. For instance, GPT-3.5-Turbo does not show good understanding of Hausa, a language native to Nigeria as [1] shows that it only scores 6.1% on ARC-Easy-Hausa (i.e. worse than a random guess of 25%).
>
> 3. While our research team is multicultural, we are not able to read major native languages used in all of the regions we are interested in. Because a major consideration in our data curation is to ensure high quality of the dataset, we needed to manually vet all responses. However, this would be infeasible if samples were collected in a wide variety of languages rather than in English alone.
>
> Nonetheless, we agree with the reviewer that collecting cultural knowledge in native languages is useful as future work and will allow us to better understand how language proficiency interacts with cultural knowledge. We hope what we have done in this paper (providing a cultural dataset in English, the world’s lingua franca) can serve as a firm foundation to enable such a follow up work.
>
> Reference:
>
> [1] https://openai.com/index/gpt-4o-system-card/
>
>
> > By looking at the resource, it seems that question is generic, can be applicable to different countries by replacing country. See first example here. For the second example, answer varies location to location within the culture.
>
> In our data collection process, we provide the annotators a high degree of freedom to phrase questions and hence, it is expected that some questions will sound generic other than the country name (e.g. the first example - “In the Netherlands, which of the following is an unusual common public practice?”). We believe this is fine as it does not make questions easier for models to answer. However, it’s important to highlight that other questions will not be generic (e.g. “In Hong Kong culture, how do you signal to the server that you would like more water added to your tea while dining in local Chinese restaurants?”.) Changing the region name in such questions to other names (such as India or Japan) will not apply.
>
> We also agree that there are certain cultural questions in which the answer will vary from location to location within a region, especially for larger countries with diverse “subcultures’. To reduce the occurrence of such questions, we asked five annotators to independently validate the correctness of the answer. Only when there is agreement between at least 4 annotators do we include the question into our benchmark. This reduces the likelihood that answers with substantial intra-cultural variation will make it into our dataset. Looking more closely into the second example - “In Korean dining etiquette, what is a common practice regarding drinks and paying for the meal?”, “Younger people pour drinks for the elders and the elders pay for the meal.” is a consensus answer that all five annotators agreed on.
>
> > What does it mean to be "less represented", as per as online digital content representation languages are typically categorized as low, medium and high resource languages.
>
> We avoid using the terms (low, medium and high resource languages) to prevent potential misunderstanding that our dataset is multilingual (as it is only based in English). In its replacement, we use the term “less-represented” in direct contrast with the term “over-represented” in the sense that there are certain cultures (e.g. United States) which are “over-represented” in LLM training data, following other works [1] and [2].
>
> Reference:
>
> [1] https://arxiv.org/abs/2203.13722 Probing Pre-Trained Language Models for Cross-Cultural Differences in Values
>
> [2] https://arxiv.org/abs/2311.14096 Cultural Bias and Cultural Alignment of Large Language Models

---

> ### Author Response · Authors · 2024-11-23
> **Response to Reviewer M6Pe – (5/5)**
>
> > What is the purpose of CULTURALBENCH-Hard?
>
> We propose CulturalBench-Hard to reduce the likelihood that the model is able to answer the question correctly by guessing. In the CulturalBench-Easy setting, the model is given a multiple-choice question with four options, with one correct answer - meaning the chance of getting the answer with just a random guess will be 25%. Furthermore, if models simply use the heuristic of guessing the option that is semantically closest to the name of the culture (e.g. Hong Kong) without even knowing what the question is, this chance rises to 40%.
>
> To make the questions more challenging, we made a simple change to question format to prevent the ‘choose semantically closest option’ guessing strategy and lower the chance of getting the answer right due to random guesses. Specifically, we separately asked the model whether each option is True or False as an answer to the question. Because the model never sees all options at once, it cannot use ‘choose semantically closest option’ as a strategy. Furthermore, we only consider that a model is able to get a question right if it is able to provide the expected answer for all four options. This brings the chance of getting the question right due to random guesses to 0.5^4, which is 6.25%.
>
> Our approach of making CulturalBench-Hard challenging for LLM is effective because the best performing LLM (GPT-4o) only achieves 61.5% compared to human performance of 92.4%.
>
> > Which model is used for brainstorming?
>
> We used GPT-3.5-Turbo to provide near real-time interactions with human annotators, rather than to keep humans waiting for too long, which can lead them to be disengaged.
>
> > "culturally relevant scenarios" --> Was it completely open? Were there any instructions like what it means to be culturally relevant scenarios? Example in Figure 10 is not clear. As an annotator how should one start with it?
>
> Yes, it was completely open. We provided minimal guidance to annotators by only guiding them to consider ‘culturally relevant scenarios’ as what make their culture different from the United States and gave examples relating to social traditions, behaviors, customs and norms. One example of a culturally relevant scenario is ‘Celebrating a girl’s 15th Birthday in Mexico’. During our annotation process, we provided an URL to a web platform to annotators giving them instructions and showing them examples before giving them an opportunity to interact with the system. For anonymity reasons, we are unable to provide the URL in this response. However, across more than 400 annotators who had a chance to use our platform, all of them were able to start the annotation quickly upon receiving the URL, which made us believe that the learning curve for our platform to be fairly gentle.
>
> > Answers often come in different forms such short or long for the same question. Did you observe any such? How did you manage that? It may come from multiple annotators. What values (e.g., temperature) did you choose on the LLM while generating question and answer? Was it same for all annotators.
>
> The variation in answer length is expected because we gave full flexibility for human annotators to edit the options as they wish. We do not consider this variation to be an issue and instead value the diversity of option lengths. Each sample (including the main question as well as the four options) is created by only one annotator, though working collaboratively with AI. We used a temperature of 0.3 together with the default top p of 1 for all question/option-generation LLM calls as this balances creativity and groundedness well. In generating the question, we restrict max output tokens to 100 and in generating the option, we restrict it to 50. In practice, we find that this has been consistently sufficient as the max token length has never been reached (i.e. output is never truncated).
>
> > L348: For the evaluation, did the model always generated correct answer? It is possible that it produced extra tokens. How did you deal with that?
>
> Models were able to generate answers in the correct format (e.g. A, B, C or D for CulturalBench-Easy and True or False for CulturalBench-Hard) most of the time (>98%) and did not generate extra tokens when we set max output token = 2. To ensure fair comparison between models, we use a strict evaluation criteria: the model answer is only considered correct when the output only contains the correct label (e.g. A for CulturalBench-Easy). See more details in Section 6.

---

> ### Comment · Reviewer_M6Pe · 2024-11-30
>
> Authors clarified most of my concerns, though some are still not convincing. I raised my overall assessment to 5.

---

> > ### Author Response · Authors · 2024-12-01
> > **Response to comment by Reviewer M6Pe**
> >
> > Thank you for your response. Can the reviewer elaborate on the specific parts of the reviewer’s concerns that have not been convincingly clarified? We are happy to add further experiments, analysis and writing to further elucidate these parts.

---

### Meta-Review · Area_Chair_2D2n · 2024-12-22

**Metareview:**

This paper proposes a question and answer dataset that aims to assess cultural knowledge (for 45 global regions including the underrepresented ones like Bangladesh, Zimbabwe, and Peru) of LLMs. The paper also uses this dataset to assess the cultural knowledge of off-the-shelf open and closed LLMs and identifies issues, such as lack of knowledge about certain regions. The paper targets an important problem that hasn't been of sufficient interest previously. Weaknesses identified by the reviewers include limited positioning of the work with respect to previous benchmarks and lack of clarity on decisions and experiments.

**Additional Comments On Reviewer Discussion:**

There were several discussions between the reviewers and the authors, these helped clarifying several of the reviewer questions. However, there are still some issues and it would be good to re-review the paper after the suggestions are integrated.

---

### Decision · Program_Chairs · 2025-01-22

Reject